# Advancing SVD-based LLM Compression via Layer-Wise Error Model Search

Moritz Thoma [1 2]   Maximilian Groezinger [2 *]   Maximilian Forstenhäusler [2 3 *]   Emad Aghajanzadeh [1 2]
Manoj Rohit Vemparala [2]   Christos Anagnostopoulos [3]   Pierpaolo Mori [2]   Nael Fasfous [2]
Alexander Frickenstein [2]   Daniel Mueller-Gritschneder [4]   Ulf Schlichtmann [1]

## Abstract

Low-rank SVD-based compression offers a powerful strategy to reduce the computational costs of LLMs. However, existing methods face two key limitations: (i) global rank allocation, where uncalibrated error proxies fail to capture complex error propagation, and (ii) decomposition quality, where Fisher-based estimators suffer from severe rank collapse. In this work, we address these limitations by introducing **Layer-wise Error Modeling Search (LEMS)** and **KFAC-SVD**. LEMS advances rank allocation by introducing a layer-wise error surrogate that integrates local and global layer importance alongside a propagation bias, enabling effective global rank allocation via an ILP formulation. KFAC-SVD improves decomposition quality by utilizing token-wise statistics, mitigating the rank deficiency observed in prior Fisher-based SVD approaches. Across Mistral, Qwen3, and Llama 3 model families, we show that LEMS consistently outperforms existing search strategies, delivering significant zero-shot accuracy gains of up to 4.8 p.p. that generalize to model sizes of 70B parameters, while KFAC-SVD achieves an average perplexity improvement of 15%. *Project Page & Code: https://lems-svd.github.io*

## 1. Introduction

The rapid scaling of LLMs (Yang et al., 2025; Grattafiori et al., 2024; Jiang et al., 2023) has enabled previously unattainable capabilities, but introduced substantial computational and memory requirements that limit deployment in resource-constrained environments. To address these lim-

itations, low-rank approximation using SVD has emerged as a principled and effective compression strategy (Qinsi et al., 2025; Wang et al., 2025b;d;c; Chekalina et al., 2025). By decomposing dense weight matrices into smaller factors, SVD substantially reduces parameter counts and computational complexity while retaining easy-to-accelerate dense tensor computations. This ensures immediate compatibility with standard inference engines, avoiding the specialized hardware or sparse kernels that are often required by semi-structured or unstructured pruning approaches. Prior work has demonstrated that SVD-based methods yield significant inference acceleration, particularly in memory-bound regimes, while achieving performance competitive with state-of-the-art pruning techniques (Qinsi et al., 2025; Wang et al., 2025b;d). They also demonstrate that low-rank approximation is orthogonal to quantization, allowing the two methods to be combined effectively for maximum efficiency.

However, the full potential of SVD in modern LLMs is limited by two fundamental challenges: **global rank allocation** and **decomposition quality**. The primary bottleneck is the global allocation problem, which requires determining the optimal rank for each layer under a fixed parameter budget. Existing strategies fail to solve this optimization effectively, as they rely on uncalibrated local sensitivity proxies (Azizi et al., 2025; Wang et al., 2025b) or expensive measurements (Thoma et al., 2025; Yuan et al., 2024) to determine layer importance. These methods typically neglect the nonlinear *error explosion* inherent in deep networks (Huang et al., 2025), creating a critical misalignment between error proxy and actual degradation. Consequently, their ability to identify high-performance rank allocations remains limited. Further, high decomposition quality requires distinguishing and preserving the weight directions that are most critical for the downstream task. While Fisher-weighted SVD theoretically addresses this by aligning reconstruction error with the loss landscape (curvature), existing work (Chekalina et al., 2025) suffers from rank deficiencies under reasonable sample counts, limiting its effectiveness.

In this work, we address both challenges by introducing **LEMS** for global rank allocation and **KFAC-SVD** for robust decomposition. LEMS introduces a layer-wise error model that integrates local sensitivity, empirical calibra-

---

[*]Equal contribution  [1]Technical University of Munich, Germany  [2]BMW Group, Germany  [3]University of Glasgow, United Kingdom  [4]TU Wien, Austria. Correspondence to: Moritz Thoma <moritz.thoma@tum.de>.

*Proceedings of the 43$^{rd}$ International Conference on Machine Learning*, Seoul, South Korea. PMLR 306, 2026. Copyright 2026 by the author(s).

tion, and a bias correcting error propagation. By explicitly modeling layer-wise error accumulation, we can formulate the search as an ILP and solve it efficiently. KFAC-SVD mitigates the rank deficiency that limits the decomposition quality of prior Fisher-based SVD by transitioning from sequence-based to token-wise statistics for the Fisher approximation. By accumulating token-level loss curvature information, we effectively prevent rank collapse, yielding a decomposition of significantly higher fidelity. Our contributions are threefold:

1. **LEMS.** We propose a layer-wise error model for rank allocation in SVD that captures both local sensitivity and global error propagation modeling error propagation via a bias. To obtain global rank allocations we search the error model with an ILP formulation.

2. **Robust Fisher Decomposition (KFAC-SVD).** We leverage a token-wise Kronecker-factor estimation in SVD to mitigate the rank bottleneck of prior Fisher-based SVD and improve its decomposition quality.

3. **Empirical Validation.** Across Mistral, Qwen3, and Llama 3, **KFAC-SVD** improves WikiText2 perplexity by an average of **15**% over baselines. **LEMS** further drives substantial accuracy gains of **1.7**-**4.8** p.p. compared to the best existing rank allocation methods. Crucially, it also generalizes to activation-based SVD and scales to the 70B parameter regime, achieving a **4.0** p.p. improvement over uniform rank allocation.

## 2. Related Work

**Low Rank Decomposition.** Standard SVD compression replaces a weight, $\mathbf{W}$, with a rank-$r$ approximation, $\mathbf{W}_r$, that minimizes $\|\mathbf{W} - \mathbf{W}_r\|_F$. This is optimal for weight reconstruction, but often suboptimal for model performance, as deviation magnitude is a poor proxy for importance. To better match the forward computation, recent methods optimize an **activation-based objective**, targeting the layer-output error $\|\mathbf{WX} - \mathbf{W}_r\mathbf{X}\|_F$. ASVD (Yuan et al., 2024) weights the rows of the weight matrices by activation importance. Building on this idea, SVD-LLM (Wang et al., 2025d) demonstrates that whitening $\mathbf{W}$ using a Cholesky factor of the input activation covariance yields the optimal rank-$r$ approximation for minimizing the Frobenius layer-output error. Subsequent work improves numerical stability (Wang et al., 2025b) and estimation accuracy of the covariance (Thoma et al., 2025) while keeping the same output-reconstruction objective. Basis-Sharing (Wang et al., 2025a) builds on SVD-LLM but reuses low-rank bases across layers to reduce parameters on disk, creating unique memory-computation trade-offs. DOBI-SVD (Qinsi et al., 2025) departs from direct weight whitening by truncating activations and fitting low-rank weights that best preserve the retained features: Because activations are modified during the forward pass,

the effective objective differs slightly from SVD-LLM-style methods. It also introduces remapping, which is essentially quantization of decomposed components. BitStack (Wang et al., 2025c) fundamentally alters the decomposition process, separating it from prior SVD-based methods.

While activation-based methods effectively preserve feature representations, complementary research targets the loss landscape directly. These methods model compression impact via a local second-order (Taylor) approximation, yielding **Fisher-weighted objectives**. Fisher-weighted SVD (Hsu et al., 2022) uses a diagonal Fisher approximation to weight truncation by parameter importance, while GFWSVD (Chekalina et al., 2025) improves curvature modeling using a Kronecker-factored Fisher, demonstrating that two-sided whitening yields the optimal rank-$r$ solution under this approximation.

*Our approach.* Activation-based methods are robust but loss-agnostic, whereas Fisher-weighted compression is loss-aware. Current Fisher-based work relies on sequence-gradient fitting, sample-limited at LLM scale. To mitigate, we estimate Kronecker factors from token-wise second moments, making truncation viable and stable in large models.

**Searching for Ranks.** Assigning different ranks to different layers can substantially improve quality at a fixed compression budget (Yuan et al., 2024; Thoma et al., 2025; Wang et al., 2025b; Azizi et al., 2025). Existing approaches primarily differ in their search and optimization strategies.

**(i) Greedy rank allocation.** Several methods use greedy procedures that progressively increase compression while tracking reconstruction metrics. Both MRCS (Azizi et al., 2025) and FastLRD (Luo et al., 2024) iteratively adjust ranks to control energy loss or weight-based similarity, aiming to keep layer-wise reconstruction error small.

**(ii) Balanced reconstruction error.** A commonly adopted heuristic allocates compression across layers to achieve similar reconstruction errors. ASVD evaluates candidate layer-wise configurations and applies a binary search to find a uniform error threshold that satisfies a global budget. Flar-SVD (Thoma et al., 2025) performs per-layer binary searches to more precisely equalize reconstruction error in vision transformers, while SVD-LLMv2 (Wang et al., 2025b) groups weights by type (attention vs. MLP) and redistributes compression within each group based on relative loss contributions inferred from removed singular values.

**(iii) Learned rank allocation.** A further class of methods learns rank importance directly. ComCat (Xiao et al., 2023) introduces trainable signals for per-layer rank allocation, while ARS (Gao et al., 2024b) and the search component of DOBI-SVD (Qinsi et al., 2025) optimize ranks via gradient-updated masks that are controlled by a small hyper-network.

**(iv) Further sparsity allocation.** Related search problems appear in structured and semi-structured pruning. MDP (Sun et al., 2025) formulates structured sparsity allocation as an ILP, enabling one-shot global optimization. ATP (Huang et al., 2025) analyzes error propagation and shows that lower bounds on reconstruction error increase for later layers when early layers are compressed, motivating a simple linear allocation scheme that increases compression depth-wise.

*Our approach.* Current rank-search methods often (i) rely on greedy updates with limited global optimality guarantees, (ii) target uniform error distributions rather than minimizing total error, or (iii) require gradient-based optimization with tuned hyper parameters. Inspired by ILP-based sparsity allocation (Sun et al., 2025) and propagation analyses (Huang et al., 2025), we propose to directly **minimize total error** under a global budget by solving an ILP over discrete per-layer rank candidates using a layer-wise proxy model.

## 3. Preliminaries

Current SVD-compression methods generally optimize either a task-aware importance metric or an activation-preserving reconstruction error. In **Fisher-weighted SVD-compression**, we seek a rank-$r$ approximation, $\widehat{\mathbf{W}}$, for a linear layer weight, $\mathbf{W} \in \mathbb{R}^{d_{\text{out}} \times d_{\text{in}}}$, that minimizes the compression-induced loss increase, $\Delta\mathcal{L}$. By using a second-order Taylor expansion and approximating the Hessian with the Fisher matrix, $\mathbf{F}$, (Martens, 2020) we get the objective:

$$\min_{\text{rank}(\widehat{\mathbf{W}}) \leq r} \Delta\boldsymbol{\theta}^{\top} \mathbf{F} \, \Delta\boldsymbol{\theta} \tag{1}$$

where $\Delta\boldsymbol{\theta} = \text{vec}(\mathbf{W} - \widehat{\mathbf{W}})$ represents the vectorized change in parameters. Since the complete Fisher matrix, $\mathbf{F} \in \mathbb{R}^{(d_{\text{out}} d_{\text{in}}) \times (d_{\text{out}} d_{\text{in}})}$, is intractable due to the dimensionality of LLM layers, i.e., $(d_{\text{out}} d_{\text{in}}) \times (d_{\text{out}} d_{\text{in}})$, structured approximations are imperative. Thus, we employ Kronecker-factored approximations of the layer-wise Fisher matrix,

$$\mathbf{F} \approx \mathbf{A} \otimes \mathbf{G}, \qquad \mathbf{A} \in \mathbb{R}^{d_{\text{in}} \times d_{\text{in}}}, \ \mathbf{G} \in \mathbb{R}^{d_{\text{out}} \times d_{\text{out}}}, \tag{2}$$

where $\mathbf{A}$ and $\mathbf{G}$ are positive semidefinite factors. Substituting Equation (2) into Equation (1) yields a Frobenius-norm objective in whitened coordinates:

$$\begin{aligned} &\min_{\text{rank}(\widehat{\mathbf{W}}) \leq r} \Delta\boldsymbol{\theta}^{\top} (\mathbf{A} \otimes \mathbf{G}) \, \Delta\boldsymbol{\theta} \\ &= \min_{\text{rank}(\widehat{\mathbf{W}}) \leq r} \left\| \mathbf{G}^{1/2} \left( \mathbf{W} - \widehat{\mathbf{W}} \right) \mathbf{A}^{1/2} \right\|_F^2. \end{aligned} \tag{3}$$

Equation (3) generalizes the **activation-based SVD** whitening used in SVD-LLM (Wang et al., 2025d). Specifically, setting $\mathbf{G} = \mathbf{I}$ recovers the SVD-LLM objective, reducing the problem to minimizing the layer-wise output reconstruction error weighted solely by the input covariance, $\mathbf{A}$.

Then the optimal rank-$r$ approximation for Equation (3) is obtained by truncating the SVD of $\widetilde{\mathbf{W}} = \mathbf{G}^{1/2}\mathbf{W}\mathbf{A}^{1/2}$ (Chekalina et al., 2025). Specifically, let $\widetilde{\mathbf{W}}_r = \mathbf{U}_r \boldsymbol{\Sigma}_r \mathbf{V}_r^{\top}$ denote the rank-$r$ truncation of $\widetilde{\mathbf{W}}$, then

$$\widehat{\mathbf{W}} = \mathbf{G}^{-1/2} \, \mathbf{U}_r \boldsymbol{\Sigma}_r \mathbf{V}_r^{\top} \, \mathbf{A}^{-1/2}. \tag{4}$$

$\widehat{\mathbf{W}}$ is implemented as two consecutive linear layers by symmetrically splitting the singular values,

$$\mathbf{B} = \boldsymbol{\Sigma}_r^{1/2} \mathbf{V}_r^{\top} \mathbf{A}^{-1/2}, \qquad \mathbf{C} = \mathbf{G}^{-1/2} \mathbf{U}_r \boldsymbol{\Sigma}_r^{1/2}, \tag{5}$$

such that $\widehat{\mathbf{W}} = \mathbf{C}\mathbf{B}$ and the forward pass becomes $y = \mathbf{C}(\mathbf{B}x)$. The bias terms, if present, are kept unchanged.

## 4. Methodology

We present **KFAC-SVD** that uses a token-wise estimation to improve decomposition in Fisher-SVD, and **LEMS**, an error-model-driven strategy for global rank allocation.

### 4.1. Decoupled Token-Wise Estimation (KFAC-SVD)

**Motivation.** Once the Kronecker factors, $\mathbf{A}$ and $\mathbf{G}$, are fixed, the Fisher-weighted truncation in Equation (3) has a standard closed-form solution and is identical across methods. GFWSVD (Chekalina et al., 2025) determines the factors $\mathbf{A}$ and $\mathbf{G}$ using $k$ per-sequence gradients (individual ones or sums over micro-batches). Let $\mathbf{G}_i^{(\mathbf{W})} = \nabla_{\mathbf{W}} \ell_i \in \mathbb{R}^{d_{\text{out}} \times d_{\text{in}}}$ denote the weight gradient for sequence $i$ and $\boldsymbol{g}_i = \text{vec}(\mathbf{G}_i^{(\mathbf{W})})$ its vectorized form, then the empirical Fisher $\widehat{\mathbf{F}}$ is formalized as

$$\widehat{\mathbf{F}} = \frac{1}{k} \sum_{i=1}^{k} \boldsymbol{g}_i \boldsymbol{g}_i^{\top}. \tag{6}$$

By fitting a rank-1 approximation to $\widehat{\mathbf{F}}$ in a rearranged space using Lanczos (Lanczos, 1950), GFWSVD creates a Kronecker product $\mathbf{A} \otimes \mathbf{G}$. However, as Equation (6) is a sum of $k$ outer products, it obeys the strict rank bound

$$\text{rank}(\widehat{\mathbf{F}}) \leq k, \tag{7}$$

inducing a **rank and conditioning bottleneck**. In modern LLM layers, the parameter dimension $d_{\text{out}} d_{\text{in}}$ is orders of magnitude larger than feasible $k$ (e.g., $k \leq 256$), so curvature is observed through a low-dimensional subspace. Empirically, this manifests as spectral collapse of the fitted factors (low effective rank/poor conditioning), and increased Fisher degradation after rank-$r$ truncations under Equation (3).

**Method.** To avoid the rank bottleneck of GFWSVD's per-sequence gradient-based estimator, we estimate Kronecker factors from *token-wise* second moments. For a linear layer with input $\boldsymbol{x}$ and output gradient $\boldsymbol{\delta} = \partial\mathcal{L}/\partial\boldsymbol{y}$, the per-token

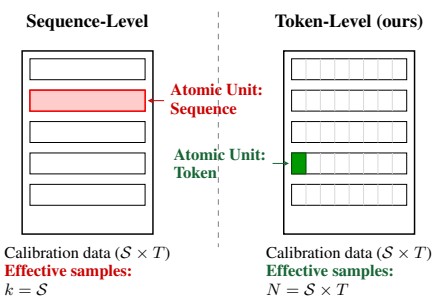

(a) **KFAC-SVD**: Per token processing to increase the effective sample count.

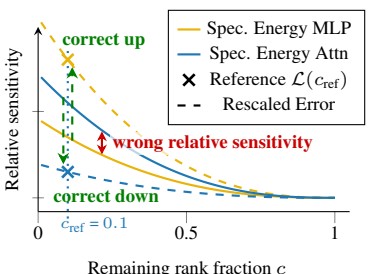

(b) **LEMS scaling**: Reference based energy rescaling to correct relative ordering.

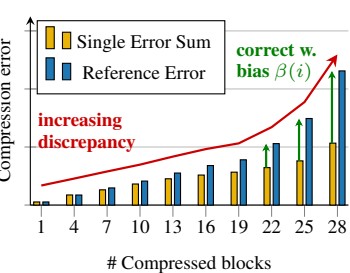

(c) **LEMS bias**: Error bias to account for accumulating error properties.

*Figure 1.* **Overview of KFAC-SVD and LEMS. (a)** KFAC-SVD mitigates the rank bottleneck of prior Fisher-based SVD by leveraging token-wise statistics (N≫k). **(b, c)** LEMS enables effective allocation by **(b)** scaling raw spectral (spec.) energy to reflect true empirical importance and **(c)** adding a depth-dependent bias to model error depth-dependent propagation. See Supp. B.1 for empirical validation.

weight gradient is $\nabla_{\mathbf{W}}\mathcal{L} = \boldsymbol{\delta}\boldsymbol{x}^{\top}$. Following (Martens & Grosse, 2015), we approximate the expectation of the Kronecker product by the Kronecker product of expectations:

$$\mathbf{F} = \mathbb{E}[(\boldsymbol{x}\otimes\boldsymbol{\delta})(\boldsymbol{x}\otimes\boldsymbol{\delta})^{\top}] \approx \mathbb{E}[\boldsymbol{x}\boldsymbol{x}^{\top}]\otimes\mathbb{E}[\boldsymbol{\delta}\boldsymbol{\delta}^{\top}] = \mathbf{A}\otimes\mathbf{G}. \quad (8)$$

Unlike fitting from a small set of full weight gradients $g_i$, KFAC estimates $\mathbf{A}$ and $\mathbf{G}$ from $N$ token instances. With $\mathcal{S}$ sequences of length $T$, the effective sample count is $N \approx \mathcal{S}T$, e.g., $256 \times 2048 \approx 5.2 \times 10^5$. We compute

$$\widehat{\mathbf{G}} = \frac{1}{N}\sum_{i=1}^{N}\boldsymbol{\delta}_i\boldsymbol{\delta}_i^{\top}, \qquad \widehat{\mathbf{A}} = \frac{1}{N}\sum_{i=1}^{N}\boldsymbol{x}_i\boldsymbol{x}_i^{\top}. \quad (9)$$

These estimators satisfy

$$\text{rank}(\widehat{\mathbf{A}}) \leq \min(d_{\text{in}}, N), \quad \text{rank}(\widehat{\mathbf{G}}) \leq \min(d_{\text{out}}, N), \quad (10)$$

therefore avoiding the $k$-limited rank bottleneck of Equation (7) and maintaining more diverse curvature directions. Although this token-wise approximation omits cross-token second moments (e.g., induced by weight sharing (Eschenhagen et al., 2023)), this trade-off proves favorable in practice, as it eliminates spectral collapse, i.e., the dominant failure mode limiting GFWSVD's performance. The full algorithm is provided in the supplementary Section C.

### 4.2. Global Rank Allocation via Layer-wise Error Modeling (LEMS)

**Motivation.** Effective rank allocation requires accurately predicting the contribution of a given rank choice, $r_i$, for a linear layer weight, e.g., $\mathbf{W}_{\text{Q}}$, $\mathbf{W}_{\text{down}}$, to the global model degradation. This problem poses three distinct challenges:

1. **Granularity (Shape):** Solving the allocation problem requires loss estimators for *all* candidate ranks across weights; however, evaluating these via forward passes is computationally intractable.

2. **Sensitivity (Scale):** Standard proxies derived from SVD spectra, e.g., tail energy, are informative but uncalibrated. A 10% spectral energy loss in a MLP projection may impact the output differently than a 10% loss in an attention projection (see Figure 1b).

3. **Propagation (Bias):** Losses are not strictly additive. Errors introduced in early layers propagate and amplify through the depth of the network. Simply summing layer-wise errors systematically underestimates global degradation for compressed models (see Figure 1c).

Existing approaches fail to address these challenges simultaneously. Methods relying on spectral energy (Wang et al., 2025b; Azizi et al., 2025) are uncalibrated (sensitivity) and ignore propagation. Similarly, measurement approaches (Yuan et al., 2024; Thoma et al., 2025) are not only intractable for fine-grained search (granularity) but also fail to model propagation. The only exception addressing propagation, (Huang et al., 2025), sacrifices sensitivity by enforcing uniform block-wise compression rates.

**Method.** To systematically address these challenges, we rely on two core components: (i) a decomposable cost model that disentangles complex loss contributions via spectral shape (for granularity), a calibration point (for scale), and a depth-dependent bias (for propagation); and (ii) an ILP that identifies the globally optimal allocation under this proxy. This enables rank selection to be formulated as a constrained combinatorial optimization problem over the set of linear layers, $\mathbf{W} = \{\mathbf{W}_1, ..., \mathbf{W}_{N_L}\}$, distributed across $B$ transformer blocks. The objective is to minimize the estimated global degradation subject to a parameter budget $P_{\text{target}}$, which is determined by the target retention rate $c_{\text{target}}$.

**(i) Reliable Error Model.** Our goal is to disentangle the complex, non-linear loss landscape into tractably additive terms. We approximate the coupled end-to-end loss using a *layer-additive* surrogate, $L_i(c)$. We define this cost as

the product of three calibrated terms, mapping a retained parameter ratio, $c$, to an estimated global degradation:

$$L_i(c) = \underbrace{S_i(c)}_{\text{Shape}} \cdot \underbrace{\alpha_i}_{\text{Scale}} \cdot \underbrace{(1 + \beta(d_i))}_{\text{Bias}}. \quad (11)$$

Here, $d_i \in \{1, \ldots, B\}$ represents the block depth where layer $i$ resides.

**(a) Analytical Sensitivity (Shape).** Whether using Fisher-weighted or activation-based SVD, the singular value spectrum $\{\sigma_{i,k}\}$ quantifies the local information retained at any specific rank. We exploit this to handle the granularity challenge. Let $\widetilde{\mathbf{W}}_i$ be the whitened weight matrix (as defined in Section 3). We define the normalized tail energy $S_i(c)$ for a parameter ratio $c = r_i \frac{d_{\text{in}} + d_{\text{out}}}{d_{\text{in}} d_{\text{out}}}$ (ratio of retained parameters relative to full matrix) as:

$$S_i(c) = \frac{\sum_{k > r_i} \sigma_{i,k}^2}{\sum_k \sigma_{i,k}^2} = \frac{\|\widetilde{\mathbf{W}}_i - \widetilde{\mathbf{W}}_{i,r_i}\|_F^2}{\|\widetilde{\mathbf{W}}_i\|_F^2}. \quad (12)$$

$S_i(c)$ yields a computationally cheap, high-resolution curve describing the *relative shape* of degradation. It quantifies the rate at which reconstruction error grows as rank decreases.

**(b) Empirical Calibration (Scale).** The spectral measure $S_i(c)$ captures the *relative* quality of ranks within a layer, but its magnitude is hard to compare between layer (types) (Wang et al., 2025b). To fix the scale, we measure the model degradation (KL divergence between original and compressed model output) at a single reference point $c_{\text{ref}}$ (e.g., 0.1) for each layer in isolation. We derive a scaling factor, $\alpha_i$, to map the spectral shape to the task loss:

$$l_i(c) = S_i(c) \cdot \underbrace{\left( \frac{\mathcal{L}_{\text{KL},i}(c_{\text{ref}})}{S_i(c_{\text{ref}}) + \varepsilon} \right)}_{\alpha_i}. \quad (13)$$

This combines the best of both: the dense granularity of the analytical spectrum and the ground-truth accuracy of empirical probing, as illustrated in Figure 1b.

**(c) Interaction Correction (Bias).** Summing local costs, $l_i(c)$, ignores that early errors can amplify during propagation (Huang et al., 2025). As illustrated in Figure 1c, this mismatch grows with depth, motivating a depth-dependent prior. However, rigid block-level constraints or Lagrangian relaxations (Bragin et al., 2015) are ill-suited here, as they conflate individual layer importance through score aggregation and complicate discrete (non-smooth) ILP penalty multiplier tuning. Instead, we apply a *soft* bias that modulates layer costs by block index, $d_i$, modeling cumulative downstream impact as a harmonic tail sum:

$$\beta(d_i) = \kappa \cdot \frac{\sum_{k=d_i}^B k^{-\gamma}}{Z}, \quad Z = \sum_{k=1}^B k^{-\gamma}. \quad (14)$$

This profile assigns larger penalties to early blocks and decays smoothly with depth. Intuitively, $\beta(\cdot)$ provides a flexible relaxation of the depth-dependent prior without imposing rigid monotonicity. $\kappa$ and $\gamma$ are calibrated via lightweight Tree-structured parzen estimation (TPE), adapting them to architectural properties. See Supplement B.1.1 for additional details).

**(ii) Integer linear program (ILP).** Given the calibrated costs for all layers, $L_i(c)$, we select exactly one rank configuration $j \in \mathcal{J}_i$ for each linear layer $i \in \{1, \ldots, N_L\}$ to minimize the total cost under a budget constraint. Let $p_{ij} = P_{\text{full}}^{(i)} \cdot c_{ij}$ denote the retained parameters and let $x_{ij} \in \{0, 1\}$ indicate whether the configuration is selected. Then the optimization problem is formulated as:

$$\min_{x_{ij}} \quad \sum_{i=1}^{N_L} \sum_{j \in \mathcal{J}_i} L_i(c_{ij}) \cdot x_{ij} \quad (15)$$

$$\text{s.t.} \quad \sum_{j \in \mathcal{J}_i} x_{ij} = 1, \quad \forall i \in 1, \ldots, N_L$$

$$\sum_{i=1}^{N_L} \sum_{j \in \mathcal{J}i} p_{ij} \cdot x_{ij} \leq P_{\text{target}}.$$

This allows to identify the globally optimal allocation for our proxy objective in seconds using off-the-shelf solvers.

## 5. Evaluation

We evaluate our proposed methods across diverse LLM families, SVD baselines and perform exhaustive ablation studies. Additional results, evaluation and further information are available in the supplementary material.

**Models, Baselines and Datasets.** We compare LEMS to a variation of searches. Specifically, we pick one representative search from each group mentioned: greedy, MRCS (Azizi et al., 2025); similar error, ASVD (Yuan et al., 2024) and SVD-LLMv2 (Wang et al., 2025b)); gradient based, ARS (Gao et al., 2024b); and the principled ATP (Huang et al., 2025) search. Additionally, we compare our method against a diverse set of recent SVD-based approaches, namely FWSVD (Hsu et al., 2022), ASVD (Yuan et al., 2024), SVD-LLM (Wang et al., 2025d), SVD-LLMv2 (Wang et al., 2025b), DOBI-SVD (Qinsi et al., 2025) and GFWSVD (Chekalina et al., 2025). We evaluate Mistral (Jiang et al., 2023), Llama 3 (Grattafiori et al., 2024), and Qwen3 (Yang et al., 2025) models on Wiki-Text2 (Merity et al., 2017), PTB (Marcus et al., 1993), C4 (Raffel et al., 2020) and six zero-shot benchmarks (PIQA (Bisk et al., 2020), OpenBookQA (Mihaylov et al., 2018), HellaSwag (Zellers et al., 2019), ARC-Easy/ARC-Challenge (Clark et al., 2018), and Winogrande (Sakaguchi et al., 2021)), using the LM-Eval library (Gao et al., 2024a).

*Table 1.* Comparison of **LEMS** against state-of-the-art search baselines across modern LLMs and two compression rates.

| | Search Method | Mistral-7B | | | | Llama 3-8B | | | | Qwen3-8B | | | |
|---|---|---|---|---|---|---|---|---|---|---|---|---|---|
| | | Wiki ↓ | C4 ↓ | PTB ↓ | Acc ↑ | Wiki ↓ | C4 ↓ | PTB ↓ | Acc ↑ | Wiki ↓ | C4 ↓ | PTB ↓ | Acc ↑ |
| | Baseline | 5.25 | 8.10 | 27.72 | 63.95 | 6.14 | 9.47 | 9.90 | 63.34 | 9.71 | 15.52 | 15.43 | 62.03 |
| 0.8 | **uniform** | 7.14 | 19.43 | 64.84 | 52.35 | 11.44 | 52.00 | 80.06 | 47.69 | 12.52 | 31.85 | 47.25 | 53.56 |
| | ASVD | 7.20 | 22.41 | 78.51 | 47.81 | 12.92 | 63.83 | 99.64 | 45.95 | 15.70 | 43.79 | 72.61 | 47.72 |
| | SVD-LLMv2 | 7.13 | 19.35 | 64.84 | 52.68 | 11.40 | 52.00 | 80.38 | 47.70 | 12.52 | 31.72 | 47.44 | 53.36 |
| | MRCS | 7.11 | 19.13 | 68.21 | 51.75 | 11.99 | 55.89 | 83.91 | 46.63 | 14.52 | 37.38 | 53.75 | 52.04 |
| | ARS | 7.26 | 11.74 | 47.71 | 54.58 | 11.81 | 20.44 | 28.83 | 54.31 | 11.58 | 22.41 | 25.05 | 55.66 |
| | ATP | 7.14 | 19.43 | 64.84 | 52.35 | 11.07 | 35.32 | 81.32 | 51.56 | 12.52 | 31.85 | 47.25 | 53.56 |
| | **LEMS** | **5.98** | **11.24** | **37.09** | **57.67** | **8.16** | **17.52** | **22.80** | **55.99** | **10.38** | **18.32** | **18.32** | **59.28** |
| 0.6 | **uniform** | 14.38 | 72.05 | 208.48 | 39.10 | 48.56 | 475.36 | 1312.50 | 34.39 | 21.68 | 101.61 | 195.85 | 39.63 |
| | ASVD | 16.78 | 102.00 | 317.89 | 35.56 | 75.21 | 571.15 | 1577.00 | 33.98 | 29.22 | 122.56 | 228.97 | 36.09 |
| | SVD-LLMv2 | 14.90 | 73.18 | 209.30 | 39.29 | 47.53 | 467.99 | 1242.65 | 34.43 | 21.42 | 96.95 | 189.82 | 39.74 |
| | MRCS | 14.21 | 74.05 | 253.45 | 37.44 | 67.95 | 610.37 | 1815.12 | 33.41 | 38.72 | 161.10 | 322.90 | 36.42 |
| | ARS | 19.43 | 37.09 | 166.87 | 40.83 | 28.77 | **56.55** | **112.03** | 40.57 | 29.51 | 83.91 | 161.10 | 40.72 |
| | ATP | 13.59 | 60.43 | 189.08 | 40.24 | 25.14 | 173.51 | 467.99 | 39.95 | 19.47 | 84.89 | 186.88 | 40.20 |
| | **LEMS** | **10.58** | **34.37** | **124.00** | **45.60** | **17.86** | 81.01 | 201.28 | **43.09** | **15.70** | **57.22** | **120.66** | **45.51** |

**Implementation Details.** To ensure fair comparison, all methods are tested under identical settings (incl. seed). For calibration of SVD algorithms, we use 256 random sequences of length 2048 from WikiText2. For search methods involving data, we use 32 samples from WikiText2 following ASVD. For the ILP search we use the Gurobi solver (Gurobi Optimization, LLC, 2024), with a per layer search range of $c_i = [0.3, 1.0]$. Compression ratios refer to the remaining parameters of linear layers (excluding head). All experiments are run on a single NVIDIA H200 GPU.

**Additional Results.** For detailed tables, data ablations, more architectures and results on non-standard SVD (i.e. BasisSharing, BitStack) we refer to the Supplement.

### 5.1. Rank Allocation Results

We evaluate the proposed **LEMS** rank allocation against representative searches. All methods employ the *same* decomposition pipeline, calibration data and candidate ranks, differing only in rank selection (refer to Supplement A). We report perplexity on WikiText2, C4 and PTB (lower is better), and average zero-shot accuracy (higher is better) for two compression ratios (0.8, 0.6) across three architectures (Table 1). Per-task accuracies are provided in Supplement D.

*Uniform search*: Across all six settings, LEMS achieves the highest accuracy and substantially reduces perplexity relative to uniform rank allocation. The accuracy gain over the *second-best* method ranges from **1.68** to **4.79** percentage points (p.p.) while the (mean) perplexity averaged over WikiText2, C4 and PTB is reduced, relative to uniform allocation, by **39.34** to **83.65%**. *Baseline searches*: Among the baselines, ASVD and SVD-LLMv2 frequently fail to outperform uniform allocation, consistent with the limitations of noisy, sensitivity proxies (cf. Supplement B.1.2) and

constraints that enforce overly similar compression across heterogeneous layer types (cf. Supplement B.1.3), respectively. Similarly, MRCS struggles in most settings. ARS is the strongest baseline, achieving second-best performance in both accuracy and perplexity across most settings, yet it requires gradients, more data, and hyperparameter tuning to reliably meet target ratios. Despite its simplicity, ATP, which is the only baseline explicitly modeling accumulating errors, generally matches or narrowly outperforms uniform allocation and remains competitive with other searches, especially on Llama 3. However, because it assumes error accumulation as a linear function, it cannot capture layer- and type-specific importance deviations. Overall, the results strongly validate the efficacy of LEMS.

### 5.2. SVD-Compression Results

We evaluate **KFAC-SVD** against state-of-the-art methods across three LLM families: Mistral, Llama 3, and Qwen3. Table 2 details the results for all approaches under uniform compression rates (0.9, 0.7) and calibration sample count, with an additional line for our search-enhanced variant. KFAC-SVD consistently yields the lowest perplexity, with gains most pronounced on Llama 3. At a 0.7 compression ratio, KFAC-SVD improves perplexity on WikiText2 by **38%** over the best baseline (DOBI-SVD), extending to **65%** with LEMS. Zero-shot accuracy largely follows this trend: under mild compression (0.9), KFAC-SVD avoids the sharp accuracy drop seen in several baselines and stays in the mid-**50**s, while LEMS recovers near-lossless performance (**62.0%** vs. **63.3%** original). Strictly comparing to prior Fisher-based methods, KFAC-SVD outperforms the baselines by up to **7.2** p.p. *without* the addition of LEMS. GFWSVD degrades significantly under identical conditions, reaffirming its conditioning challenges. The only excep-

*Table 2.* Comparison of **KFAC-SVD** with activation- and Fisher-based SVD across modern LLMs under uniform compression and **LEMS**.

| | SVD Method | Mistral-7B | | | | Llama 3-8B | | | | Qwen3-8B | | | |
|---|---|---|---|---|---|---|---|---|---|---|---|---|---|
| | | Wiki ↓ | C4 ↓ | PTB ↓ | Acc↑ | Wiki ↓ | C4 ↓ | PTB ↓ | Acc ↑ | Wiki ↓ | C4 ↓ | PTB ↓ | Acc ↑ |
| | Baseline | 5.25 | 8.10 | 27.72 | 63.96 | 6.14 | 9.47 | 9.90 | 63.33 | 9.71 | 15.52 | 15.43 | 62.04 |
| 0.9 | FWSVD | 9.47 | 13.75 | 58.35 | 56.93 | 42.11 | 59.96 | 69.83 | 48.88 | 16.95 | 24.75 | 31.54 | 53.82 |
| | ASVD | 9.14 | 13.41 | 49.13 | 57.30 | 65.09 | 77.60 | 207.67 | 47.89 | 20.36 | 29.98 | 36.37 | 51.19 |
| | SVD-LLM | 6.46 | 15.07 | 66.12 | 56.81 | 10.14 | 39.63 | 49.91 | 52.27 | 12.52 | 27.40 | 36.66 | 55.96 |
| | SVD-LLMv2 | 6.46 | 15.07 | 66.12 | 56.78 | 10.18 | 39.71 | 50.40 | 52.17 | 12.52 | 27.45 | 36.80 | 55.80 |
| | DOBI-SVD | 7.11 | 15.19 | 63.58 | 55.17 | 11.22 | 34.77 | 42.36 | 52.98 | 13.46 | 28.88 | 35.81 | 55.48 |
| | GFWSVD | 31.66 | 65.09 | 305.72 | 40.73 | 2569.75 | 3827.63 | 8491.96 | 33.40 | 171.49 | 350.51 | 765.57 | 46.66 |
| | **KFAC-SVD** | 6.22 | 13.10 | 44.04 | 56.92 | 8.84 | 25.69 | 32.60 | 54.45 | 11.51 | 23.76 | 28.00 | 57.07 |
| | **+LEMS** | **5.37** | **8.49** | **29.86** | **62.77** | **6.58** | **11.11** | **11.29** | **61.99** | **9.85** | **16.04** | **15.77** | **62.89** |
| 0.7 | FWSVD | 34.84 | 50.30 | 215.10 | 44.24 | 716.39 | 837.54 | 1939.75 | 33.40 | 41.37 | 68.75 | 90.72 | 42.68 |
| | ASVD | 28.16 | 42.94 | 163.64 | 46.09 | 10989.41 | 9399.74 | 36032.89 | 33.03 | 70.66 | 96.95 | 131.49 | 43.10 |
| | SVD-LLM | 10.96 | 42.60 | 169.49 | 44.39 | 34.64 | 376.04 | 600.91 | 37.69 | 17.11 | 63.09 | 136.19 | 45.76 |
| | SVD-LLMv2 | 10.96 | 42.60 | 169.49 | 44.35 | 34.98 | 378.99 | 615.16 | 37.70 | 17.15 | 63.33 | 136.72 | 45.62 |
| | DOBI-SVD | 12.37 | 39.71 | 170.82 | 42.94 | 31.54 | 500.12 | 223.67 | 38.48 | 18.54 | 63.83 | 94.34 | 46.24 |
| | GFWSVD | 2549.75 | 3404.36 | 5355.81 | 32.29 | 41798.89 | 53670.83 | 72222.00 | 31.77 | 21684.98 | 27627.38 | 50814.52 | 34.06 |
| | **KFAC-SVD** | 9.29 | 34.64 | 110.72 | 45.76 | 19.43 | 151.34 | 253.45 | 40.73 | 14.69 | 50.00 | 93.97 | 46.78 |
| | **+LEMS** | **7.42** | **18.25** | **67.95** | **51.52** | **11.07** | **33.51** | **65.09** | **49.39** | **11.81** | **28.33** | **37.97** | **53.80** |

tion is Mistral, where baselines like ASVD achieve higher zero-shot accuracy despite worse perplexity. This implies Mistral relies on features well-aligned with activation magnitude, favoring simpler proxies in uniform settings. Overall, KFAC-SVD provides the most effective decomposition, especially when compared with other Fisher-based methods, outperforming state-of-the-art baselines in the vast majority of evaluated settings.

## 5.3. Ablations

This section investigates the source of KFAC-SVD and LEMS performance gains through empirical validation.

### 5.3.1. SCALABILITY OF LEMS

To assess the scalability of LEMS, we evaluate the Qwen3 family (4B-32B) across five compression ratios (0.5-0.9). To demonstrate that LEMS is agnostic to a particular decomposition objective, we report results for both KFAC-SVD and the activation-based SVD-LLM. Figure 2 contrasts uniform allocation (solid) to LEMS (dashed), with numerical results provided in Supplement D.

Across all budgets and model sizes, LEMS consistently improves both perplexity and downstream utility. Averaged over ratios and models, it reduces perplexity by **23%** for KFAC-SVD and **34%** for SVD-LLM, while increasing zero-shot accuracy by **+8.3** and **+9.1** p.p., respectively. Gains are largest under aggressive compression: at 0.5, LEMS cuts perplexity by ∼**47%** (KFAC-SVD) / ∼**57%** (SVD-LLM) on average and recovers **+6.3** / **+8.7** p.p. accuracy. Crucially, it also resolves the sharp accuracy dips observed in Qwen3-32B under uniform allocation for both decompositions. The benefits extend beyond Qwen3: on the substantially larger Llama 3.1-70B, LEMS reduces WikiText2

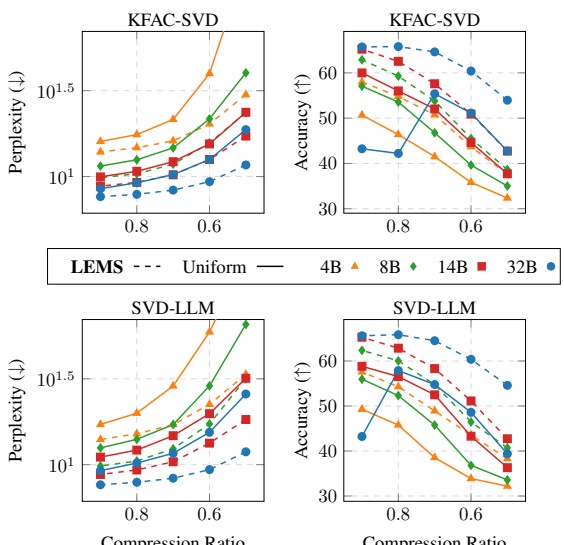

*Figure 2.* WikiText2 perplexity (log scale) and average accuracy of KFAC-SVD and SVD-LLM on Qwen3 models with uniform compression and LEMS across multiple compression ratios (0.5-0.9) and a range of sizes 4B-32B.

perplexity from 7.9 to **4.5** and increases accuracy from 62.6 to **66.6** (see Supplement B.4). Comparing decompositions methods, KFAC-SVD achieves approximately **15%** lower perplexity than SVD-LLM under uniform allocation, demonstrating stronger alignment with the downstream loss. When combined with LEMS, perplexities become tightly matched between the two methods, with SVD-LLM retaining a marginal edge in zero-shot accuracy (∼0.1 p.p.). Notably, the larger gains observed for SVD-LLM with LEMS suggest that suboptimal rank allocation is often a primary source of degradation under uniform compression. Differently, through the calibrated layer-wise error model and global ILP optimization, LEMS overcomes the degradation

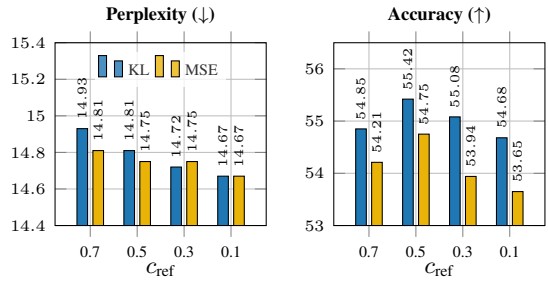

*Figure 3.* WikiText2 perplexity and zero-shot accuracy comparison of KL (blue bars) and MSE (yellow bars) as rescale importance metrics with different $c_{ref}$ values on Qwen3-4B.

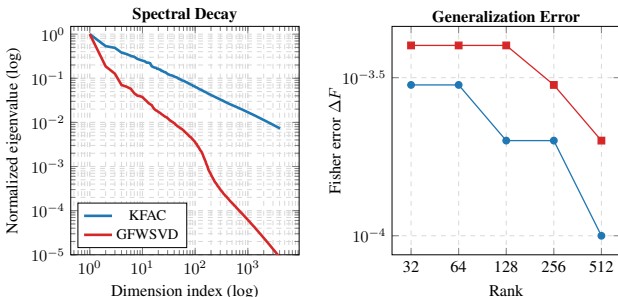

*Figure 4.* Spectral decay of Lanczos Fisher approximation (GFWSVD) compared to our KFAC approximation (left) and resulting fisher error on unseen gradients (right), showing lower spectral decay and better generalization error for KFAC.

of uniform compression yielding consistent gains.

### 5.3.2. INDIVIDUAL CONTRIBUTIONS TO LEMS

To validate components of our rank allocation strategy, we evaluate LEMS's error modeling components on Llama 3-8B and Qwen3-8B (Table 3). Introducing ILP search via spectral energy, $S_i$, reduces perplexity by roughly 18% for Llama 3 and 11% for Qwen3, adding a modest 1 minute to the runtime over the baseline. Incorporating KL-based calibration ($l_i$) yields further reductions of about 7% for both Llama 3 and Qwen3. Although this adds 17 to 22 minutes, the scores are fully cacheable and reusable across compression ratios, making this a *one-time investment*.

Finally, cross-layer bias correction, $\beta$, incurs negligible overhead (2 minutes) yet yields a 6% reduction for Llama 3. While gains are marginal for Qwen3, this behavior highlights the method's adaptivity: the corrections scale with error accumulation, acting aggressively for sensitive models while remaining neutral for resilient ones (see Supplement B.1.1). Moreover, the compression of 0.8 used here is modest, limiting the propagation effects. At 0.6, the effect becomes more pronounced, reducing the perplexity of Llama 3-8B by over 50% compared to the baseline without it. Overall, LEMS requires 20-25 minutes for the initial run and only **3 minutes** for subsequent runs, on the evaluated 8B-parameter architectures. An extended analysis about the performance contributions of global optimization and data-driven bias calibration is provided in Supplement Section B.1.4.

*Table 3.* WikiText2 performance and added execution time of individual parts of our method on Llama 3-8B and Qwen3-8B.

| **LEMS** components | Llama 3-8B | | Qwen3-8B | |
| --- | --- | --- | --- | --- |
| | Wiki ↓ | time ↓ | Wiki ↓ | time ↓ |
| SVD (uniform) | 11.44 | **start** | 12.52 | **start** |
| + ILP search (energy) | 9.36 | +1 min | 11.16 | +1 min |
| + KL calibration | 8.69 | +17 min | 10.40 | +22 min |
| + cross layer bias | **8.16** | +2 min | **10.34** | +2 min |
| search time (w/wo cache) | **3 min** / 20 min | | **3 min** / 25 min | |

### 5.3.3. LEMS ERROR CALIBRATION

We investigate the impact of the calibration point $c_{ref}$ used to estimate sensitivity during the search. Figure 3 compares KL divergence vs. MSE across varying $c_{ref}$ on Qwen3-4B, indicating that KL divergence consistently outperforms MSE in zero shot accuracy; improving it up to 1.1 p.p. For perplexity, MSE performs slightly better than KL, but with small margins. When evaluating different $c_{ref}$ values for the calibration, we observe that calibrating in the high-compression regime ($c_{ref}$=0.1) yields slightly lower perplexity (14.7) than calibrating in the low-compression one (14.9). We hypothesize that severe constraints isolate the parameters most critical for preserving the data distribution. Hence, although $c_{ref}$=0.1 incurs a marginal zero-shot reduction relative to $c_{ref}$=0.3 (-0.4 p.p.), we prioritize its better generative fidelity (perplexity) and adopt it as our default configuration, see Supplement B.1.2 for more details.

### 5.3.4. KFAC-SVD ESTIMATION QUALITY

After demonstrating consistent gains in task performance of the KFAC estimation of $\widehat{A}$ and $\widehat{G}$ over GFWSVD's estimation in Table 2, we further analyze the impact of the **sample-induced rank bottleneck**. Figure 4 (left) displays the normalized eigenvalue decay of the induced Kronecker metric, $\widehat{A} \otimes \widehat{G}$, for both approaches, demonstrating that GFWSVD suffers from severe **spectral collapse**, where the eigenspectrum decays rapidly. As a result, truncation quickly discards the information curvature structure, thereby limiting generalization. This is quantified by measuring the Fisher degradation, $\Delta F$, on held-out data across truncation ranks, indicating how well the estimator captures true loss curvature on unseen sequences (see Supplement B.2 for details). Figure 4 (right) confirms that our token-wise estimator consistently achieves lower $\Delta F$ across all truncation ranks. This supports the idea that preventing spectral collapse is essential for robust generalization, a finding further supported by effective rank analysis in Supplement B.2.

# 6. Conclusion and Limitations

In this work, to enable effective rank allocation for LLMs, we propose **LEMS**, a rank allocator driven by a calibrated layer-wise error model of spectral shape, scale, and bias. LEMS demonstrates consistent gains independent of model size and across activation- and Fisher-based SVD. Additionally, we addressed the rank bottleneck of prior Fisher-based SVD by introducing **KFAC-SVD**, which estimates the Fisher matrix with a token-wise approximation, improving decomposition quality. While improving Fisher-SVD, our proposed KFAC-SVD approach still relies on gradient computation, limiting computational efficiency. Moreover, LEMS relies on a one-time layer-wise calibration step, which scales linearly with model depth. Future work might focus on formalizing analytical bounds on error propagation to further improve modeling accuracy and extending the approach to vision-language models (VLMs) and mixture-of-experts (MoE) models.

# Impact Statement

This paper presents work whose goal is to advance the field of machine learning. There are many potential societal consequences of our work, none of which we feel must be specifically highlighted here.

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

# Glossary

**ARS** Adaptive rank selection. 2, 5, 6, 14, 32, 33

**ASVD** Activation-aware singular value decomposition. 2, 5–7, 14, 17–23, 26, 31–33

**ATP** A theoretical perspective (on sparsity). 3, 5, 6, 14, 18, 20, 22, 23, 32, 33

**BasisSharing** Basis Sharing: Cross-Layer Parameter Sharing for LLM Compression. 6, 22, 24, 25

**BitStack** BitStack: Any-Size Compression of LLMs in Variable Memory Environments. 2, 6, 22–24

**C4** Colossal clean crawled corpus. 6, 7, 23, 25, 26, 31–33

**ComCat** Compression and customization of attention-based vision models. 2

**DOBI-SVD** Differentiable SVD for LLM compression. 2, 5–7, 23, 26, 31, 32

**FastLRD** Fast low rank decomposition. 2

**FLAR-SVD** Fast and latency-aware SVD. 2

**FWSVD** Fisher weighted SVD. 5, 7, 23, 26, 31, 32

**GFWSVD** Generalized Fisher weighted SVD. 2–8, 21–23, 26, 31, 32

**ILP** Integer linear program. 1–8, 16–19, 24, 30

**KFAC** Kronecker-factored approximate curvature. 8, 21, 22, 29

**KFAC-SVD** Kronecker-factored approximate curvature singular value decomposition. 1–4, 6–9, 14, 21–28, 31, 32, 34

**KL** Kullback-Leibler divergence. 5, 8, 14, 16, 18–21, 30

**LEMS** Layer-wise error model search. 1–9, 14, 18–20, 22–26, 28, 30–34

**Llama 2** Llama 2. 14, 22–25

**Llama 3** Llama 3. 1, 2, 5–8, 14–26, 31–33

**LLM** Large language model. 1–3, 5–7, 9, 14, 15, 26

**MDP** Multidimensional vision model pruning with latency constraint. 3

**Mistral** Mistral. 1, 2, 5–7, 14, 19, 21, 25, 26, 31–33

**MLP** Multi-layer perceptron. 2, 4, 17

**MoE** Mixture-of-experts. 9

**MRCS** Mixed rank compression strategy. 2, 5, 6, 14, 17, 19, 21–23, 32, 33

**MSE** Mean squared error. 8, 18

**p.p.** Percentage points. 1, 2, 6–8, 22, 24, 25

**PTB** Penn treebank. 6, 7, 23–26, 31–33

**Qwen3** Qwen3. 1, 2, 5–8, 14–22, 25–27, 31–34

**SVD** Singular value decomposition. 1–9, 14, 21–24, 26, 28, 29, 31

**SVD-LLM** SVD for LLM compression. 2, 3, 5, 7, 23, 25–27, 31, 32, 34

**SVD-LLMv2** Optimizing SVD for LLM compression. 2, 5–7, 14, 17, 22, 23, 26, 31–33

**TPE** Tree-structured parzen estimation. 5, 16, 30

**VLM** Vision-language models. 9

**WikiText2** WikiText2. 2, 5–8, 18, 21–26, 31–33

## A. Extended Experimental Setup

**Models Evaluated.** We evaluate the proposed layer-wise error model search (LEMS) and Kronecker-factored approximate curvature singular value decomposition (KFAC-SVD) compression method across a diverse set of modern open-weights large language models (LLMs), covering different parameter scales and architectural families. All models were sourced from HuggingFace, utilizing optimized implementations from Unsloth where available. For the **Qwen3 Family**, we utilize the official checkpoints for the entire scaling suite, specifically `Qwen/Qwen3-32B`, `-14B`, `-8B`, `-4B`, and `-1.7B` (Yang et al., 2025). For the **Llama 3 Family**, we evaluate Llama 3 , the very large Llama 3.1 (70B) and the lightweight Llama 3.2 variants (1B, 3B) (Grattafiori et al., 2024), alongside the established Llama 2-7B (Touvron et al., 2023), using the unrestricted Unsloth-optimized versions (e.g., `unsloth/llama-3-8b`). Finally, for **Mistral**, we evaluate the Mistral-7B-v0.1 model (Jiang et al., 2023) via `unsloth/mistral-7b`.

**Search Baselines and Implementation Details.** To ensure a fair comparison, all search methods utilize the same decomposition approach and candidate rank sets where applicable. We implemented or adapted the baselines as follows:

- **Activation-aware singular value decomposition (ASVD)** (Yuan et al., 2024): We utilize the official public codebase with default settings as reference, where the search evaluates sensitivity on the calibration set to select ranks from a discrete set of multipliers.

- **Mixed rank compression strategy (MRCS)** (Azizi et al., 2025): As no official code was available, we reimplemented the algorithm following the published methodology. Following their setup, we set the sensitivity parameter $\gamma = 80$ and limit the search to 500 iterations. To maintain a valid search space comparable to other methods, we lower-bound the compression ratio per layer at $c_i = 0.3$.

- **Optimizing SVD for LLM compression (SVD-LLMv2)** (Wang et al., 2025b): We re-implemented the search algorithm based on the paper description, as the code remains unavailable.

- **Adaptive rank selection (ARS)** (Gao et al., 2024b): We used the official ARS codebase with three crucial modifications to ensure fair benchmarking. First, we initialize the model with our pre-computed singular value decomposition (SVD) factors rather than using their decomposition implementation to isolate the benefit of the search algorithm. Second, we aligned their parameter counting and compression ratio logic with our framework to ensure all methods target the exact same parameter budget. Third, we adjusted the stability thresholds because the default values caused severe over-pruning (model collapse) at aggressive compression ratios ($0.6\times$) for Llama 3 and Mistral. For hyperparameters, we used the authors settings for Llama 3-8B across all models. In our tests, ARS typically converges within $\sim 200$ iterations using about $7\times$ the data of ASVD and LEMS.

- **A theoretical perspective (on sparsity) (ATP)** (Huang et al., 2025): We execute the grid search procedure as specified in the original paper. We perform 10 grid-search steps over the valid linearly space and report the best performing configuration.

**LEMS (Proposed):** For our method, we use the Optuna framework (Akiba et al., 2019) to fit the bias parameters. To maintain computational parity with ATP, we limit the optimization to 10 trials. The search space for the bias magnitude is bounded to $\kappa \in [0.0, 3.0]$ and the decay shape to $\gamma \in (0.0, 7.0)$. For the scale calibration, we measure the Kullback-Leibler divergence (KL) sensitivity at a reference compression of $c_{\text{ref}} = 0.1$, as the signal-to-noise ratio of the relative importance is larger under strong compression. Finally, we constrain the layer-wise compression range to match ASVD. Additional details on our methods design choices and the algorithm are provided in the supplementary Section B.1 and C, respectively.

## B. Additional Ablations

### B.1. Extended Explanations and Ablations for LEMS

This section provides empirical validation for the design choices of our LEMS algorithm. We first investigate the necessity of the bias term by quantifying error propagation in Supplement B.1.1. We then analyze the scale component in Supplement B.1.2, comparing different importance metrics and calibration points to identify the most robust signal for global rank allocation. Next, we provide a detailed comparison between LEMS, ATP, and ASVD in Section B.1.3. Finally, in Section B.1.4 we discuss how the individual components of LEMS contribute to its performance and how the combination of the components allows surpassing prior works.

### B.1.1. BIAS FUNCTION IN ERROR MODELING

**Theoretical Justification.** The necessity of a depth-dependent bias term is grounded in the theoretical analysis of error propagation. Recent work by Huang et al. (2025) formalizes the concept of *Reconstruction Error Explosion* (Theorem 3.2), proving that in a simplified linear network, increasing the error in layer $i$ strictly increases the **lower bound** of the error in layer $i + 1$:

$$\mathcal{L}(\boldsymbol{W}_{i+1}, \boldsymbol{X}_{i+1}) > \sigma_{\min}^2(\widetilde{\boldsymbol{W}}_{i+1}) \cdot \mathcal{L}(\boldsymbol{W}_i, \boldsymbol{X}_i). \tag{16}$$

This inequality implies a cascade effect where early-layer errors are amplified by the singular values of all subsequent layers. However, this theoretical bound assumes a strictly linear feed-forward structure. Modern LLMs introduce complex non-linearities (e.g., SwiGLU), normalizations (RMSNorm), and crucial residual connections that dampen or distort this propagation in ways that are analytically highly difficult to model precisely. Consequently, while theory confirms that an error penalty *must* exist and should be higher for initial layers, deriving its exact analytical form is unfeasible for Transformers. We therefore adopt a hybrid approach: we accept the theoretical mandate for a depth-dependent penalty but determine its specific *shape* and *magnitude* empirically. Our bias term $\beta(d_i)$ is thus a parameterized correction that adapts the theoretical *error explosion* principle to the empirical reality of residual-heavy LLM architectures.

**Quantifying Error Underestimation.** To verify that this theoretical error accumulation manifests in practice, despite the mitigating effects of residual connections and norm layers, we perform a sequential compression experiment. We first measure the error induced by compressing each block $d_i \in \{1, ..., B\}$ individually. We then compress the model cumulatively from the last block to the first, measuring the actual, total degradation at each step. Figure 5 compares the naive sum of individual errors against the measured total error. The results confirm a systematic underestimation: as more blocks are compressed, the true error diverges from the additive proxy. This divergence is particularly pronounced in the Llama 3.2-3B model, where early perturbations amplify significantly as they propagate. In Qwen3-4B, while the additive approximation holds better, the underestimation remains present. This confirms that simply minimizing local errors is insufficient. Hence, we find that the optimization objective needs to be corrected with a depth-dependent bias $\beta(d_i)$ to approximate the true upper bound of the loss.

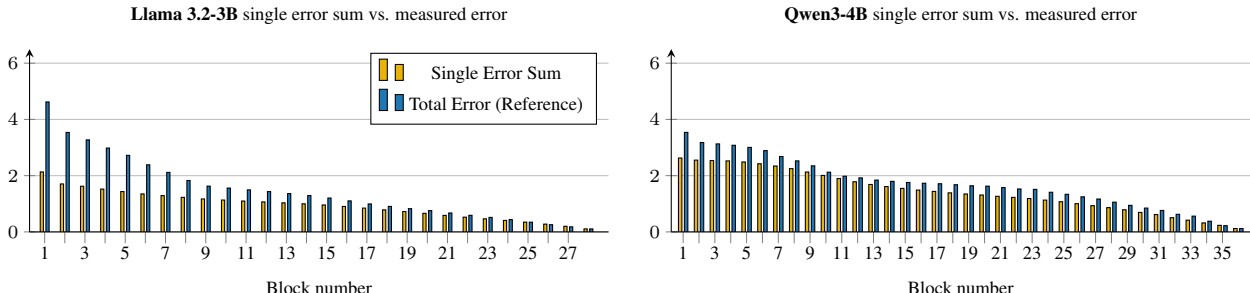

*Figure 5.* Systematic error underestimation. The yellow bars (sum of individual errors) consistently fall below the blue bars (measured total error), confirming that cross-layer interactions amplify degradation. This gap motivates the bias correction $\beta(d_i)$.

**Modeling the Bias Function.** We model the bias $\beta(d_i)$ as a depth-dependent penalty that decreases for later blocks, reflecting the observation that errors introduced early in the network have more opportunity to propagate and amplify (Huang et al., 2025). We parameterize $\beta(d_i)$ using a harmonic series controlled by a magnitude scalar $\kappa$ and a decay shape $\gamma$:

$$\beta(d_i, \kappa, \gamma) = \kappa \cdot \frac{\sum_{k=d_i}^{B} k^{-\gamma}}{\sum_{k=1}^{B} k^{-\gamma}} \tag{17}$$

Figure 6 illustrates some shapes tested, ranging from steep decay ($\gamma = 3$, heavily penalizing the early blocks) to linear decay ($\gamma = 0$). The scalar $\kappa$ ranges from 0 (no bias) to 10. Figure 7 presents the ablation results. The impact of the bias correction is significant: for Llama 3.2-3B, the best configuration ($\kappa = 8, \gamma = 0$) achieves a perplexity of **42.27**, whereas a poor choice yields **90.01**. Crucially, *any* bias correction consistently outperforms the uncorrected baseline. We observe similar trends for Qwen3-4B, where the optimal setting ($\kappa = 6, \gamma = 3$) yields a perplexity of 30.32 compared to > 40 for the uncorrected search. These results highlight that accounting for error accumulation is not merely a theoretical refinement but a prerequisite for stability in high-compression regimes.

Different bias scaling functions based on the layer depth position $d_i$: $\beta(d_i, \gamma, \kappa = 1)$

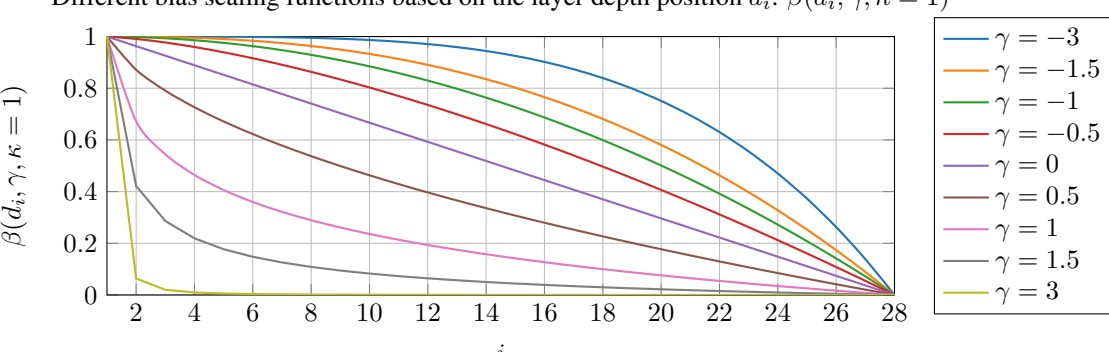

*Figure 6.* Candidate bias functions $\beta(d_i, \gamma, \kappa = 1)$. We test various decay profiles, from linear ($\gamma = 0$) to strictly exponential ($\gamma = \pm 3$), to model the error propagation penalty.

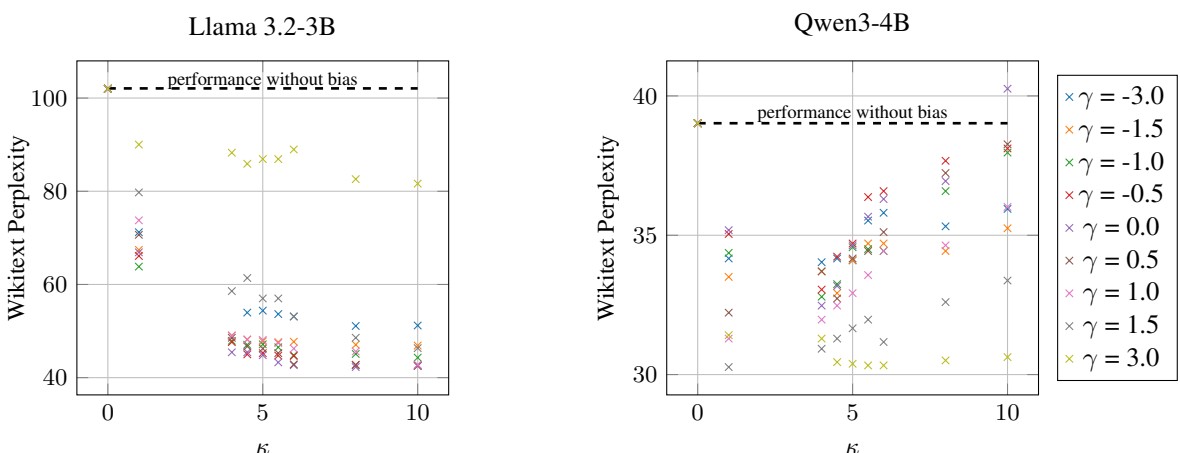

*Figure 7.* Grid Search for bias parameters $\kappa$ and $\gamma$. Scatter plots show perplexity (lower is better) for varying magnitude $\kappa$ and decay $\gamma$ for Llama 3.2-3B and Qwen3-4B. The improvements observed across both models support the benefit of assigning higher importance to earlier layers.

**Determining Bias Parameters in Practice.**    Since the exact magnitude of error propagation depends on architecture depth and residual structure, the optimal bias parameters, $\kappa$ and $\gamma$, cannot be fixed universally. To determine them efficiently, we implement a lightweight hyperparameter search using the Tree-structured parzen estimation (TPE) algorithm provided by Optuna (Akiba et al., 2019). The process is formulated as a bilevel optimization: in the outer loop, the TPE sampler suggests candidate values for $\kappa$ and $\gamma$. In the inner loop, we (1) update the cost function $L_i(c)$ with these parameters, (2) solve the integer linear program (ILP) to obtain the optimal rank configuration, (3) temporarily compress the model to these ranks, and (4) measure the KL divergence on a small calibration set. Because the ILP solver is extremely fast ($\approx$ 1s) and we utilize precomputed outputs for the uncompressed model, this entire loop is computationally inexpensive. We run 10 trials, which complete in minutes on a single GPU, yielding a fitted error propagation model that is calibrated to the specific target model and compression budget.

**Discussion - Optimization formulation.**    We note that our interaction correction term, $\beta(d_i, \kappa, \gamma)$, admits two complementary interpretations: (i) as a calibrated reweighting of the per-module surrogate costs, and (ii) as an implicit mechanism for enforcing a depth-wise structural prior supported by theory and prior empirical findings.

A classical Lagrangian formulation would operationalize this explicitly by introducing a constraint on depth-weighted compression (or degradation) in early blocks and associating a multiplier $\lambda$ that quantifies the cost of violating this prior, formalizing a surrogate Lagrangian relaxation (Bragin et al., 2015). In practice, we instead fit $\kappa$ and $\gamma$ via a lightweight bilevel loop that directly optimizes the ground-truth proxy metric (KL on a small calibration set). This approach is more robust in the discrete ILP setting: small dual-ascent updates of $\kappa$ can induce abrupt jumps in the rank allocation, and hand-specifying a constraint threshold is model- and budget-dependent. Our TPE-based calibration avoids these issues while

operationalizing the same prior, selecting bias parameters that best align the ILP solution with observed KL behavior.

Conceptually, the Lagrangian perspective remains useful: $\kappa$ can be interpreted as an implicit "shadow price" on early-block compression, and extending the method to additional depth-wise structural constraints (e.g., monotonicity or smoothness across blocks) naturally corresponds to introducing additional multipliers. Future work could explore surrogate- or level-based Lagrangian relaxations in the ILP to tighten the relaxation and further improve rank allocation search efficiency and performance (Bragin et al., 2015; Bessa et al., 2025).

### B.1.2. CORRECT CROSS-LAYER SCALING (IMPORTANCE)

**The Need for Calibrated Importance.** To allocate ranks optimally, the search algorithm must weigh the cost of compressing an multi-layer perceptron (MLP) projection layer in the first block of a model against an Attention projection layer near the model output. This requires a global importance metric. If the proxy objective wrongly assumes one layer-type is easier to compress than another, the ILP solver will aggressively truncate the sensitive layer, degrading performance.

**Challenges in Importance Estimation.** Accurate sensitivity analysis is computationally expensive. Llama 3-8B has 32 blocks with 7 linear layers each, which in turn have hundreds or even thousands of valid rank options, making the evaluation of every rank configuration intractable. Existing solutions fall into two camps:

1. **Calibration-based** (e.g., ASVD (Yuan et al., 2024)): Measures empirical loss on a validation set. While accurate, it is costly (hours per model) and sparse (checking only a few discrete ranks), often missing the optimal operating point.

2. **Energy-based** (e.g., SVD-LLMv2 and MRCS (Wang et al., 2025b; Azizi et al., 2025)): Uses the Frobenius norm of discarded weights (or activations). This is computationally inexpensive and dense (available for every rank), but uncalibrated. A 10% energy loss in layer $i$ might damage the output far more than a 10% energy loss in layer $j$.

Fisher-weighted objectives theoretically provide a more downstream (loss-aware) energy estimate. However, standard estimation only captures *local* curvature, inherently assuming fixed inputs. Therefore, they fail to model how the *distribution shift* caused by compressing layer $i$ affects the compression of layer $i+1$ (Huang et al., 2025). Moreover, the reliability of the estimation is limited by the approximation quality of the Fisher information (Kunstner et al., 2019) and may be subject to different scaling across layers due to different gradient magnitudes.

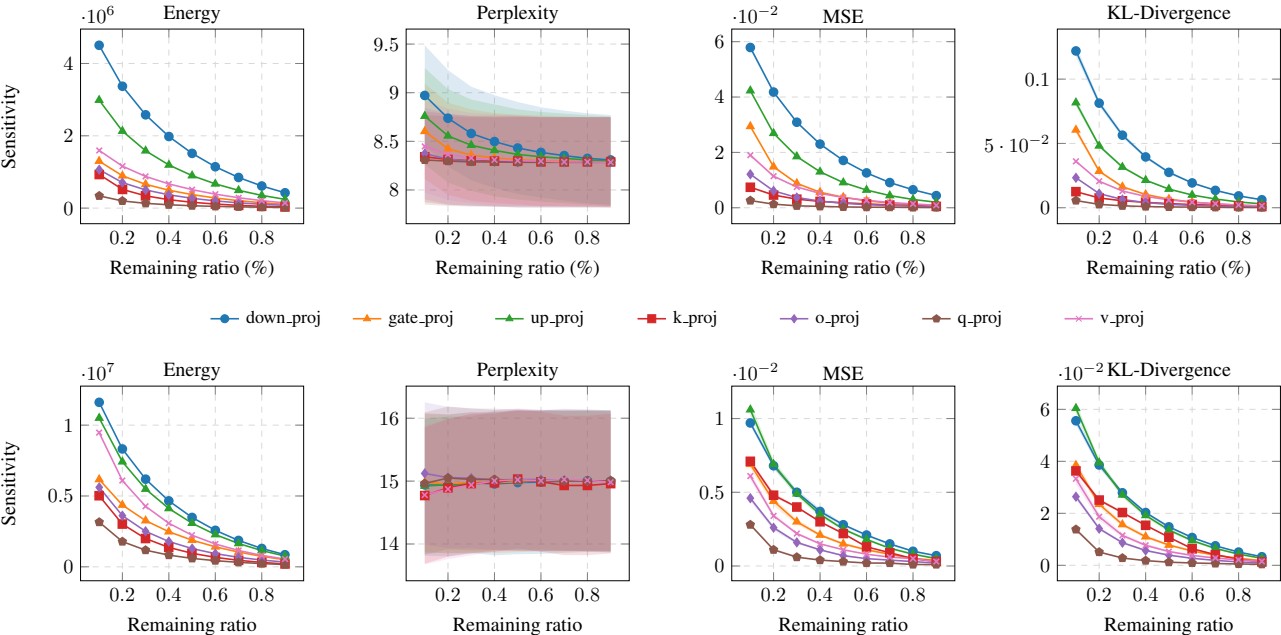

*Figure 8.* Comparison between different sensitivity assessment metrics across 9 compression ratios for all linear layers of Llama 3.2-3B (top) and Qwen3-4B (bottom) in block 15.

*Table 4.* Evaluation of different $c_{\mathrm{ref}}$ values for MSE and KL on both Llama 3.2-3B and Qwen3-4B.

| $c_{\mathrm{ref}}$ | Llama 3.2-3B | | | | Qwen3-4B | | | |
| --- | --- | --- | --- | --- | --- | --- | --- | --- |
| | Wiki ↓ | | Acc ↑ | | Wiki ↓ | | Acc ↑ | |
| | KL | MSE | KL | MSE | KL | MSE | KL | MSE |
| 0.7 | 10.26 | 10.61 | 50.11 | 47.59 | 14.93 | 14.81 | 54.85 | 54.21 |
| 0.5 | 10.08 | 10.24 | 49.75 | 48.95 | 14.81 | 14.75 | **55.42** | 54.75 |
| 0.3 | 9.87 | 10.02 | 50.33 | 49.80 | 14.72 | 14.75 | 55.08 | 53.94 |
| 0.1 | **9.81** | 9.94 | **50.60** | **50.60** | **14.67** | **14.67** | 54.68 | 53.65 |

**Selecting the Best Proxy Metric.** We aim to combine the dense resolution of the energy metric (shape) with the global accuracy of a calibration metric (scale, $\alpha_i$). We first identify the most reliable metric for this calibration. Figure 8 compares the sensitivity curves of perplexity, mean squared error (MSE), and KL-divergence across five compression ratios and four seeds on Llama 3.2-3B and Llama 3-4B. The line represents the mean while the shaded area represents standard deviation. As can be seen, **perplexity** is excessively noisy. Confidence intervals overlap significantly, making it difficult to reliably rank layer importance. Differently, **MSE and KL divergence** show stable, distinct trends that correctly differentiate layer sensitivities (e.g., correctly ranking gate_proj as more sensitive than o_proj). We select KL-divergence as it directly measures the distributional shift at the output, offering slightly better task connection than MSE, yielding in better perplexity and zero-shot accuracy results for our search tests (Table 4).

**Optimal Calibration Point.** Finally, we determine the compression ratio $c_{\mathrm{ref}}$ at which to measure this scalar importance $\alpha_i$. Table 4 shows that calibrating at $c_{\mathrm{ref}} = 0.1$ consistently yields the best downstream performance. This result aligns with signal processing principles: at $c = 0.1$ (high compression), the degradation signal is strong and clearly separable from measurement noise. At lower compression (e.g., $c = 0.9$), the degradation is subtle, and the relative importance differences between layers are masked by the noise floor of the approximation. By calibrating at the point of maximum signal fidelity ($c = 0.1$), we maximize the dynamic range of our search, allowing the ILP to distinguish layer importance even for fine-grained decisions.

### B.1.3. DETAILED COMPARISON BETWEEN LEMS, ATP AND ASVD

This section provides a detailed analysis of our pruning search strategy against two leading competitors, ATP and ASVD. To isolate the differences in their allocation strategies, we configure all methods to target 80% parameter retention on the Llama 3.2-3B model. For ASVD, we leverage our more accurate error model (without bias) to ensure the comparison focuses strictly on the search algorithm's efficacy.

An effective compression strategy must account for the heterogeneous importance of different model components. As illustrated in Figure 9, the relative importance of layers in both Llama 3.2-3B and Qwen3-4B is highly non-uniform. This variation exists across block positions/depth (Figure 9, left plots) and is especially pronounced across different layer types (Figure 9, right plots). For instance, the feed-forward layers (up-proj, down-proj) are substantially more critical than the attention projection layers (q-proj, k-proj). The primary limitation of existing methods lies in how they address this heterogeneity.

**ATP** simply employs a monotonically increasing compression ratio across blocks. As seen in Figure 10 (middle), this approach disregards the nuanced importance landscape, resulting in uniform compression across all block layer types. This indiscriminate compression leads to suboptimal performance (13.48 perplexity) due to over compression of critical layers.

**ASVD** represents an improvement by considering layer-specific importance, achieving a better perplexity of 12.06. Its compression profile is more varied than ATP's (Figure 10, bottom). However, ASVD's greedy, layer-by-layer optimization aiming towards most equal error fails to capture the global picture. While it correctly preserves more parameters in important layer types, its allocation is not as finely tuned as our LEMS, leading to less efficient parameter-performance trade-offs.

In contrast, **our** global search strategy excels by finding an optimal configuration for the given error model. The compression configuration discovered by LEMS (Figure 10, top) shows a strong correlation with the importance data from Figure 9. Moreover, its ability to perform a global trade-off based on both layer importance and absolute parameter count gives it a

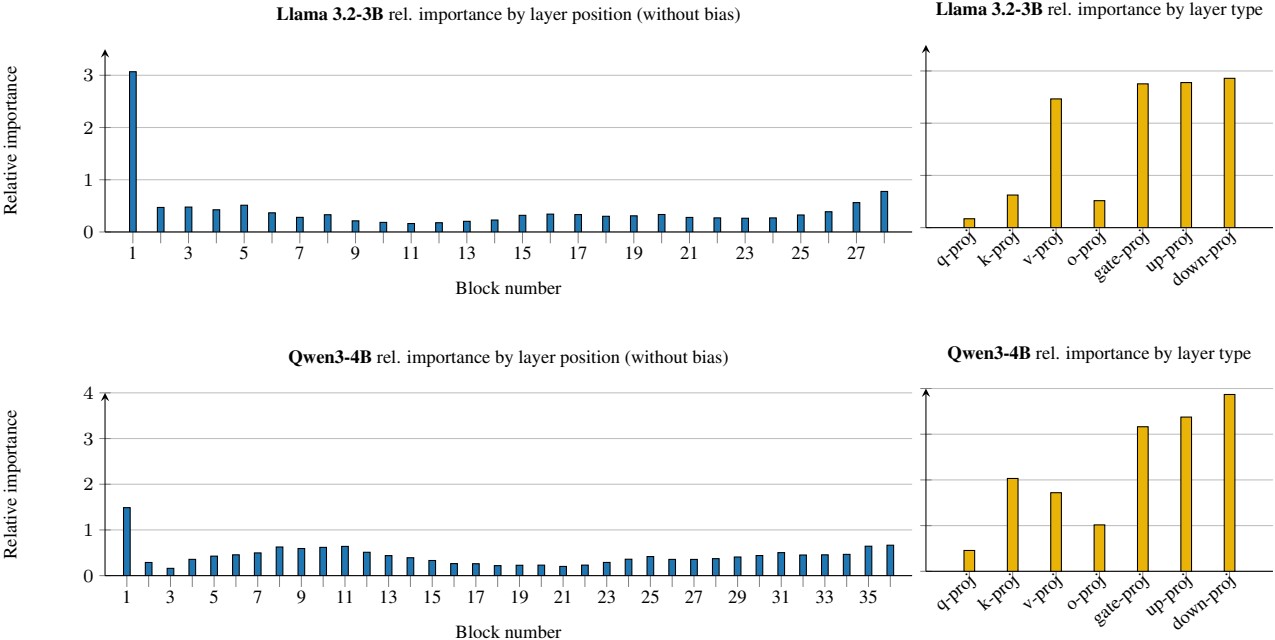

*Figure 9.* Average relative importance of block positions and layer types in Qwen3-4B and Llama 3.2-3B using KL divergence as importance metric.

major advantage. The up-proj and down-proj layers, for example, are significantly larger than the q-proj layer. Consequently, a minor increase in the compression of these large layers frees up a substantial parameter budget. Our search strategically reallocates this budget to preserve smaller, but functionally critical, layers at a much higher fidelity. This global optimization, which is beyond the scope of ASVD's greedy approach, allows our method to find a superior architectural trade-off and achieve better performance.

### B.1.4. How do Components Contribute to LEMS performance?

In the main material Section 5.3.2 we discuss incremental contributions to LEMS's performance. However, the main section does not cover how big the advantage of our global ILP solver is over using the same score with other searches (how much benefit comes from the optimization) and how much the data-based calibration of bias aids this performance. To disentangle these effects, we perform a cross-ablation on Mistral-7B, Llama 3-8B, and Qwen3-8B at 0.8 parameter retention. For this ablation, we combine each of the three score-based search procedures (MRCS, ASVD, and our ILP search) with four proxy scores: the dense spectral-energy proxy used by MRCS, the sparse KL proxy used by ASVD, our dense hybrid proxy, and our full hybrid proxy with adaptive bias. The results are presented in Table 5, except MRCS on the sparse proxy as it is not applicable to the sparse KL proxy requiring dense per-rank sensitivity curves.

The results show that neither data-based calibration nor global ILP optimization alone explains the full gain our search has over others. First, global optimization is only useful when the optimized proxy is reliable: applying our ILP search to uncalibrated energy performs poorly on Llama 3-8B, despite the solution being globally optimal under that proxy. This indicates that a global optimizer can be misled when the proxy does not reflect the true cross-layer importance. Second, proxy calibration alone is also insufficient. Retrofitting the MRCS and ASVD searches with our hybrid or hybrid+bias scores improves results for some models, but does not recover the performance of our combined method. For example, the full hybrid+bias proxy with MRCS achieves 11.47 perplexity, while the same proxy with our ILP search reaches a much better perplexity of 8.16. Consequently, the largest improvement appears only when the calibrated proxy and the global allocation through ILP are used together. The dense, rank-dependent estimate of layer impact is provided by our hybrid+bias proxy, while the ILP can globally trade off rank choices across layers under the parameter budget. This interaction yields the best perplexity and accuracy in the cross-ablation, improving from 10.18 perplexity with ASVD search and 11.47 with MRCS search to 8.16 with our ILP search. Thus, the main gain should be attributed to the alignment between proxy calibration and

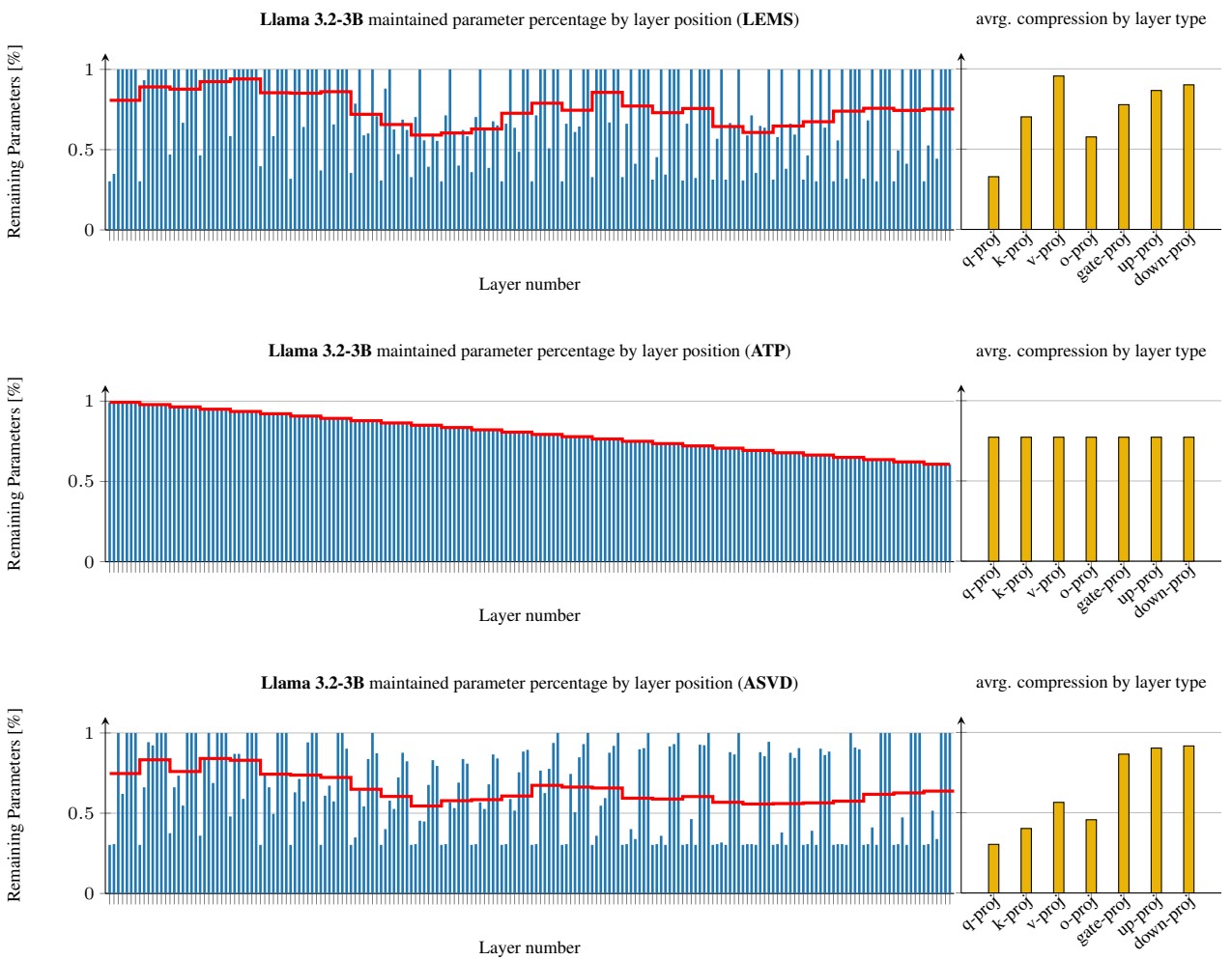

*Figure 10.* Average relative importance of block positions and block layer types in Qwen3-4B and Llama 3.2-3B using KL-divergence as importance metric.

global optimization, rather than to either component in isolation, demonstrating that the combination presented in our work is key to achieving the highest possible performance.

*Table 5.* Cross-ablation of proxy scores and search algorithms on Mistral-7B, Llama 3-8B and Qwen3-8B at 0.8 parameter retention. We report WikiText2 perplexity (lower is better) and average zero-shot accuracy in parentheses (higher is better). Best result for each proxy score is shown in bold, second best underlined. MRCS cannot use the sparse KL proxy because it requires dense per-rank sensitivity scores.

| Model | Proxy score | Type | MRCS search | ASVD search | Ours / ILP search |
|-------|-------------|------|-------------|-------------|-------------------|
| **Mistral-7B** | Energy (MRCS) | Dense | 7.11 (0.52) | **6.64 (0.55)** | 10.26 (0.48) |
| | KL (ASVD) | Sparse | n.a. | 6.85 (0.55) | **6.39 (0.55)** |
| | Hybrid (ours) | Dense | 7.22 (0.50) | 6.72 (0.52) | **6.01 (0.57)** |
| | Hybrid + bias (ours) | Dense | 7.17 (0.52) | 6.74 (0.53) | **5.98 (0.58)** |
| **Llama 3-8B** | Energy (MRCS) | Dense | 11.92 (0.47) | **10.08 (0.50)** | 14.58 (0.50) |
| | KL (ASVD) | Sparse | n.a. | **10.98 (0.49)** | 11.60 (0.49) |
| | Hybrid (ours) | Dense | 14.24 (0.46) | 12.50 (0.47) | **8.67 (0.53)** |
| | Hybrid + bias (ours) | Dense | 11.47 (0.49) | 10.18 (0.52) | **8.16 (0.56)** |
| **Qwen3-8B** | Energy (MRCS) | Dense | 14.52 (0.52) | 14.58 (0.57) | **12.52 (0.54)** |
| | KL (ASVD) | Sparse | n.a. | 11.53 (0.59) | **10.98 (0.59)** |
| | Hybrid (ours) | Dense | 13.07 (0.53) | 12.28 (0.56) | **10.40 (0.59)** |
| | Hybrid + bias (ours) | Dense | 12.79 (0.54) | 12.13 (0.56) | **10.38 (0.59)** |

## B.2. Additional Explanations and Ablations on KFAC-SVD

**Experimental Protocol and Metrics.** For the extended evaluation of KFAC-SVD, we ensure an identical data split and evaluation procedure so that everything but the factor estimation is the same. To collect token-wise activations $X \in \mathbb{R}^{B \times T \times d_{\text{in}}}$ and output gradients $\Delta \in \mathbb{R}^{B \times T \times d_{\text{out}}}$, we run forward and backward passes on our calibration data. From those, we estimate the Kronecker factors, $\widehat{\mathbf{A}}$ and $\widehat{\mathbf{G}}$, apply minimal diagonal damping to ensure Cholesky stability, and then compute the Fisher-weighted SVD truncation $\widehat{\mathbf{W}}_r$ (Equation (3)-(4)). Additionally, we obtain gradients and activations from an additional, held-out test set, that is not used during factor estimation.

To assess the estimated factors, we use two metrics. First, we measure *held-out Fisher degradation* $\Delta F$ when compressing a layer to rank $r = 128$. For that purpose, we use the unseen test set to compute the empirical Fisher quadratic form

$$\Delta F = \mathbb{E}\big[\langle \delta, (\mathbf{W} - \widehat{\mathbf{W}}_{128})x \rangle^2 \big],$$

which is measuring sequence-wise (sum over tokens per sequence, then square) Fisher deltas induced by the compression. Therefore, a lower $\Delta F$ indicates better generalization to unseen curvature directions, representing a more faithful Fisher-weighted truncation. Second, we measure the the *effective rank* $r_{\text{eff}}$ of the Kronecker spectrum $\widehat{\mathbf{A}} \otimes \widehat{\mathbf{G}}$, which is computed from the trace-normalized eigenvalues $p_i$ as $r_{\text{eff}} = \exp(-\sum_i p_i \log p_i)$. Higher $r_{\text{eff}}$ indicates more diverse directions covered.

**Extended Comparison between KFAC and Lanczos (GFWSVD).** In the main content, we illustrate the failure mode of the sequence-gradient estimator of GFWSVD (Section 5.3.4). Using the protocol outlined above, we provide broader evidence of the phenomenon across layers and models. Specifically, Table 6 reports (i) held-out Fisher degradation $\Delta F$ after rank truncation and (ii) the effective rank $r_{\text{eff}}$ of the implied Kronecker spectrum on Llama 3.2-3B and Qwen3-4B on block 0 and block 15. Across layers and blocks, Lanczos/NKP (as used in GFWSVD) frequently exhibits *spectral collapse*: the factors produce substantially lower $r_{\text{eff}}$ than the token-wise KFAC, which we attribute to the coupled estimator's reliance on a small number of sequence-level gradient samples, which concentrates curvature mass into a low-dimensional subspace. When examining the truncation quality, we find that the spectral collapse correlates with worse truncation. At rank 128, KFAC consistently achieves lower held-out Fisher degradation than GFWSVD (e.g., $\approx 1.4\times$ to $3\times$ lower $\Delta F$ depending on model/layer group in Table 6).

Combined, the results support our claim that the dominant limitation of coupled sequence-gradient fitting is the sample-limited rank bottleneck. In contrast the token-wise estimated factors of KFAC retain substantially more diverse curvature directions and generalize better to unseen gradients.

*Table 6.* Empirical Comparison of the actual Fisher information lost, when compressing with the obtained factors to rank 128, and the effective rank of each factor representation. Greater effective rank preserves more diverse directions, maintaining better generalization on the unseen fisher components.

| | | Llama 3.2-3B | | | | | | Qwen3-4B | | | | | |
| | | $\Delta F$ | | | Eff.Rank | | | $\Delta F$ | | | Eff.Rank | | |
| Idx | Layer | KFAC | GFWSVD | $\times$ | KFAC | GFWSVD | $\times$ | KFAC | GFWSVD | $\times$ | KFAC | GFWSVD | $\times$ |
|---|---|---|---|---|---|---|---|---|---|---|---|---|---|
| | q_proj | 5.4e-07 | 1.4e-06 | 0.39 | 7570.9 | 636.1 | 11.90 | 5.3e-04 | 5.6e-04 | 0.93 | 169327.1 | 2675.6 | 63.29 |
| | k_proj | 5.3e-07 | 8.0e-07 | 0.66 | 1548.4 | 545.3 | 2.84 | 8.8e-04 | 1.1e-03 | 0.81 | 67210.2 | 920.8 | 72.99 |
| | v_proj | 2.5e-05 | 5.2e-05 | 0.49 | 44295.1 | 181.1 | 244.62 | 4.6e-04 | 1.0e-03 | 0.46 | 93505.1 | 380.8 | 245.54 |
| 0 | o_proj | 4.8e-05 | 9.5e-05 | 0.51 | 255441.8 | 230.7 | 1107.10 | 5.0e-03 | 7.6e-03 | 0.66 | 259343.8 | 370.3 | 700.37 |
| | up_proj | 2.3e-04 | 3.6e-04 | 0.65 | 3133342.0 | 63.7 | 49225.85 | 8.3e-04 | 9.1e-04 | 0.91 | 2655398.0 | 1449.3 | 1832.19 |
| | down_proj | 3.0e-04 | 5.5e-04 | 0.55 | 1972431.0 | 90.1 | 21899.57 | 1.3e-03 | 2.0e-03 | 0.63 | 2969225.0 | 340.4 | 8722.68 |
| | **Avg** | **1.0e-04** | **1.8e-04** | **0.54** | **902438.2** | **291.2** | **12081.98** | **1.5e-03** | **2.2e-03** | **0.73** | **1035668.2** | **1022.9** | **1939.51** |
| | q_proj | 9.2e-06 | 2.2e-05 | 0.41 | 79847.3 | 9279.5 | 8.60 | 1.1e-04 | 3.1e-04 | 0.34 | 20170.5 | 12.4 | 1625.46 |
| | k_proj | 4.4e-06 | 9.6e-06 | 0.46 | 26065.4 | 1111.5 | 23.45 | 7.9e-05 | 1.3e-04 | 0.63 | 20775.3 | 28.3 | 733.45 |
| | v_proj | 6.6e-05 | 6.3e-05 | 1.05 | 305087.9 | 661.5 | 461.17 | 1.4e-04 | 8.5e-04 | 0.16 | 28784.4 | 10.1 | 2861.56 |
| 15 | o_proj | 7.9e-05 | 8.2e-05 | 0.97 | 673192.2 | 5528.5 | 121.77 | 3.2e-04 | 7.6e-04 | 0.42 | 178611.0 | 24.9 | 7174.04 |
| | up_proj | 1.3e-04 | 2.1e-04 | 0.63 | 2853569.0 | 4045.4 | 705.38 | 9.1e-04 | 6.1e-03 | 0.15 | 550894.1 | 12.7 | 43284.68 |
| | down_proj | 1.8e-04 | 1.9e-04 | 0.93 | 4104151.0 | 4575.7 | 896.94 | 6.0e-04 | 2.0e-03 | 0.30 | 1766736.0 | 20.9 | 84487.91 |
| | **Avg** | **7.9e-05** | **9.7e-05** | **0.74** | **1340318.8** | **4200.4** | **369.55** | **3.6e-04** | **1.7e-03** | **0.33** | **427661.9** | **18.2** | **23361.18** |

### B.3. Llama 2-7B Results

While we evaluate KFAC-SVD on a range of different relevant architectures in the main content, in this section, we want to provide additional results for Llama 2-7B (Touvron et al., 2023) as it is a very popular example in the SVD compression domain. The results are presented in Table 7 and reiterate the strong results for both KFAC-SVD and LEMS.

### B.4. Llama 3.1-70B Results

To demonstrate the scalability of LEMS beyond the Qwen3-32B architecture, we extend our evaluation to Llama 3.1-70B. As shown in Table 8, our method substantially outperforms the uniform baseline, reducing WikiText2 perplexity by 43% (7.94 to 4.53). This advantage extends to zero-shot tasks, where LEMS improves or matches the uniform baseline on every benchmark, ranging from identical performance on Winogrande to a substantial 9.0 percentage points (p.p.) gain on HellaSwag, averaging a 4.0 p.p. increase in accuracy overall. While ASVD achieves the second-best results, it incurs nearly $3.5\times$ the computational cost of LEMS and trails our accuracy by 2.5 p.p., highlighting the efficiency and effectiveness of our approach. Conversely, ATP's simple linear compression strategy (designed for large models) and SVD-LLMv2 fail to identify rank allocations with clear superiority over the uniform baseline. Similarly, MRCS struggles, falling below the uniform baseline in average accuracy while achieving the third best WikiText2 perplexity. Notably, we adjusted its hyperparameters as it required more steps to compress the larger model. Overall, the consistent performance gains seen with our method, combined with the full cacheability of sensitivity metrics make our search a compelling option for large model compression.

### B.5. Comparison with Alternative Decomposition Techniques

Besides standard layer-wise SVD-based compression we have discussed and compared with in the main material, two more recent approaches exist in related work: BitStack (Wang et al., 2025c) and BasisSharing (Wang et al., 2025a). Since both of them deviate from the other works in how the decomposition is done, in this section we explain them in more detail and compare our approach to them.

#### B.5.1. BITSTACK

**BitStack Differences to Standard SVD.** BitStack (Wang et al., 2025c) uses a fundamentally different approach to standard low-rank decomposition of weights. Specifically, they iteratively separate the sign $\mathcal{S}$ out of the weight matrix $W$ resulting in $\mathcal{S} \cdot |W|$, decompose it to SVD($|W|$), truncate the result to rank k=16 and reconstruct it to $|W|_k$. Subsequently, they compute the difference of the reconstructed and the original weight and separate sign and decompose the difference again. This way, with an increasing number of iterative steps, their weight approximation becomes better and better. As shown in Wang et al. (2025c), this method is highly effective, preserving perplexity and zero shot accuracy well, even at low memory footprints.

*Table 7.* Additional results of all methods on **Llama 2-7B** across multiple uniform compression rates and **SVD baselines**. **Bold**/Underline denote best/second-best compressed results.

|   | Method | Wiki ↓ | C4 ↓ | PTB ↓ | Openb. ↑ | ARC_e ↑ | ARC_c ↑ | WinoG. ↑ | HellaS. ↑ | PIQA ↑ | **Avg.** ↑ |
|---|--------|--------|------|-------|----------|---------|---------|----------|-----------|--------|------------|
|   | Original | 5.47 | 7.29 | 26.82 | 0.33 | 0.76 | 0.44 | 0.69 | 0.57 | 0.78 | 59.51 |
| 0.8 | FWSVD | 173.51 | 224.54 | 4138.65 | 0.15 | 0.29 | 0.21 | 0.50 | 0.28 | 0.55 | 33.10 |
|   | ASVD | 25.24 | 32.73 | 239.02 | 0.26 | 0.54 | 0.28 | 0.60 | 0.40 | 0.65 | 45.44 |
|   | SVD-LLM | 8.42 | 19.97 | 129.95 | 0.26 | 0.57 | 0.27 | 0.63 | 0.40 | 0.67 | 46.66 |
|   | DOBI-SVD | 9.58 | 20.20 | 102.00 | 0.24 | 0.52 | 0.24 | 0.60 | 0.38 | 0.64 | 43.80 |
|   | GFWSVD | 63.09 | 121.61 | 881.17 | 0.15 | 0.34 | 0.21 | 0.53 | 0.30 | 0.58 | 34.97 |
|   | **KFAC-SVD** | 7.58 | 15.95 | **70.66** | 0.26 | 0.64 | 0.30 | 0.63 | 0.42 | 0.69 | 49.18 |
|   | +LEMS | **6.19** | **9.51** | 86.23 | **0.31** | **0.72** | **0.38** | **0.68** | **0.52** | **0.75** | **56.19** |
| 0.6 | FWSVD | 699.79 | 964.00 | 5150.63 | 0.11 | 0.27 | 0.20 | 0.50 | 0.26 | 0.54 | 31.37 |
|   | ASVD | 3004.34 | 3595.72 | 3124.02 | 0.13 | 0.28 | 0.19 | 0.50 | 0.27 | 0.54 | 31.71 |
|   | SVD-LLM | 16.27 | 72.05 | 786.80 | 0.19 | 0.39 | 0.21 | 0.56 | 0.31 | 0.58 | 37.42 |
|   | DOBI-SVD | 18.50 | 60.67 | 457.14 | 0.19 | 0.34 | 0.20 | 0.53 | 0.30 | 0.57 | 35.76 |
|   | GFWSVD | 2946.23 | 3738.96 | 3680.99 | 0.14 | 0.26 | 0.22 | 0.50 | 0.26 | 0.54 | 31.88 |
|   | **KFAC-SVD** | 13.35 | 49.71 | 329.27 | 0.20 | 0.43 | 0.22 | 0.57 | 0.32 | 0.60 | 39.14 |
|   | +LEMS | **9.87** | **22.94** | **281.64** | **0.24** | **0.56** | **0.28** | **0.62** | **0.38** | **0.65** | **45.64** |

*Table 8.* Extended format zero-shot performance and perplexity of Llama 3.1-70B with a **0.8 parameter retention**.

| Method | Wiki ↓ | C4 ↓ | PTB ↓ | Openb. ↑ | ARC_e ↑ | ARC_c ↑ | WinoG. ↑ | HellaS. ↑ | PIQA ↑ | **Avg.** ↑ | time [h] ↓ |
|--------|--------|------|-------|----------|---------|---------|----------|-----------|--------|------------|------------|
| Original | 2.81 | 7.65 | 7.22 | 0.37 | 0.60 | 0.85 | 0.82 | 0.67 | 0.83 | 68.95 | - |
| uniform | 7.94 | 21.42 | 21.68 | 0.35 | 0.81 | 0.52 | **0.78** | 0.53 | 0.78 | 62.61 | - |
| ASVD | 7.33 | 26.25 | **18.72** | 0.36 | 0.83 | 0.54 | 0.75 | 0.57 | 0.79 | 64.11 | 22.5 |
| SVD-LLMv2 | 7.82 | 21.13 | 21.46 | 0.34 | 0.81 | 0.52 | 0.77 | 0.53 | 0.78 | 62.61 | <0.1 |
| MRCS | 7.52 | 22.02 | 22.23 | 0.35 | 0.81 | 0.51 | 0.76 | 0.53 | 0.78 | 62.38 | <0.1 |
| ATP | 7.94 | 21.42 | 21.68 | 0.35 | 0.81 | 0.52 | **0.78** | 0.53 | 0.78 | 62.61 | 0.2 |
| **LEMS** | **4.53** | **18.50** | 18.98 | **0.37** | **0.85** | **0.58** | **0.78** | **0.62** | **0.81** | **66.60** | 6.6 |

However, there are two main drawbacks: first, the approach is reconstructing the full weight before the forward pass, which makes it computationally expensive and results in slowdowns compared to the baseline, even with highly optimized kernels. In their paper they report it to be about 10% slower than the fp16 baseline at approximately 25% remaining size; second, separating out the binary mask makes combining it with quantization ineffective, specifically, in every iterative step a binary tensor with the size of the original weight is extracted. Hence, when using only 4 iterative steps, which is very few already, the binary masks alone take up the same amount of space as the original weight when quantized to 4 bit (excluding scaling factors). While the decomposed weight also contributes to the memory footprint and is quantizable, the space occupied by the non-quantizable binary mask is significantly larger. Hence, different to other SVD approaches, applying quantization to BitStack has negligible memory advantages. Looking at a Llama 3-8b example, the q_proj layer has a shape of $4096 \times 4096$. With just 4 iterative blocks BitStack uses $4096 \times 4096 \times 4 \times 1\,\text{bit} = 67,108,864$ for the sign mask, while using 16 ranks for each of the 4 blocks for the decomposed weight results in $(4096 \times 4 \times 16 + 4096 \times 4 \times 16) \times 16\,\text{bit} = 8,388,608\,\text{bit}$. In total this makes $75,497,472\,\text{bit}$, which is 28.125% of the original weight. Now, when considering 4 bit quantization with 128 group size, the total bit count for the decomposed part comes down from $8,388,608$ to $2,162,688\,\text{bit}$. Considering this quantization, the total comes down to 25.81% of the original weight size. Meanwhile, when considering the same weight compressed with other SVD methods and combined with quantization, **all** layer components are quantizable. Hence, when performing mild SVD compression that removes only 10% of parameters and perform 4 bit quantization on top of it, we get $(4096 \times 1844 + 1844 \times 4096) \times 16\,\text{bit} = 241,696,768\,\text{bit}$ for the decomposition, and $62,312,448\,\text{bit}$ when quantizing to 128 groups 4 bit, reducing it to 23.21% of the original size. Hence, the total reduction achieved when combining it with quantization is similar while likely achieving runtime acceleration. Therefore, overall, combining traditional SVD with quantization is more effective as it allows for inference acceleration and high weight reduction, which BitStack's sign separation approach cannot offer.

*Table 9.* Perplexity on WikiText2 (↓) and zero shot accuracy of Llama 3-8B. For BitStack we report the quantized memory, but float accuracy due to compatibility issues, favoring its results.

| Method | Weight Memory | Accuracy |
|---|---|---|
| BitStack | 5.19 GB | 59.93 |
| KFAC-SVD | **5.07 GB** | **60.83** |

**BitStack Performance Evaluation.** In this section, we compare KFAC-SVD with BitStack (Wang et al., 2025c), which we treat separately due to its fundamentally different decomposition. Unlike the other SVD-based methods evaluated so far, which preserve separate low-rank layers at inference time (Equation (5)), BitStack requires reconstructing full weights prior to computation with limited quantization impact as described above. In Table 9, we compare the memory–accuracy trade-off on Llama 3-8B, as provided in the BitStack repository. To reflect realistic deployment, we report performance under AWQ quantization for both methods. For BitStack, however, we only report the *theoretical* footprint under AWQ *without actually compressing it*, since its low-rank structure is incompatible with standard quantization kernels. Even under this favorable assumption for the baseline, our decomposition and search yield a better trade-off, improving compression while maintaining higher accuracy.

### B.5.2. BASISSHARING

**BasisSharing Differences to Standard SVD.** Standard SVD compresses every layer individually into two separate sublayers (see Equation (5)), with both of the factors being specific to that layer. Differently, BasisSharing (Wang et al., 2025a) groups adjacent layers of the same type, and shares one low-rank component between the layers of that group. Considering a group of layers $\mathcal{G} = \{i_1, \ldots, i_m\}$, members are first concatenated, then jointly compressed and finally split again into a shared and individual component

$$[\widehat{\mathbf{W}}_{i_1}, \ldots, \widehat{\mathbf{W}}_{i_m}] = \mathbf{C}_{\mathcal{G}}[\mathbf{B}_{i_1}, \ldots, \mathbf{B}_{i_m}], \qquad \widehat{\mathbf{W}}_i = \mathbf{C}_{\mathcal{G}}\mathbf{B}_i. \tag{18}$$

Since $\mathbf{C}_{\mathcal{G}}$ is only stored once per group and just $\mathbf{B}_i$ is layer specific, the number of stored parameters compared to independent SVD is reduced. Specifically, for $m$ layers of the same type with shared rank $r_{\mathcal{G}}$, independent SVD stores $mr_{\mathcal{G}}(d_{\text{out}} + d_{\text{in}})$ parameters, whereas BasisSharing only stores $r_{\mathcal{G}}d_{\text{out}} + mr_{\mathcal{G}}d_{\text{in}}$. However, while the sharing results in a lower storage footprint, the shared basis are still *applied* independently during the model inference. Hence, the computational load of the approach is higher than the one of traditional SVD at the same number of parameters.

**BasisSharing Evaluation.** In this paragraph, we compare KFAC-SVD to BasisSharing and demonstrate applicability of LEMS to BasisSharing. For the comparison, we fully reimplement BasisSharing in our repository. We further extend our LEMS formulation to work with the concept of layer sharing. To do that, our ILP formulation (Equation (15)) must be amended to ensure, that all layers inside a sharing group $\mathcal{G}$ select the same rank candidate. For that we additionally impose the constraint

$$x_{ij} = x_{i'j}, \qquad \forall i, i' \in \mathcal{G}, \quad \forall j, \quad \forall \mathcal{G}, \tag{19}$$

which forces the same intermediate rank on all layers within a group.

Starting with the assessment of the reimplementation, as shown in Table 10, when using the same 32-bit float model, dataloader and seed, our performance matches the reference (Tab. 2 in (Wang et al., 2025a)), validating our implementation. However, for the rest of the comparisons we opt to use the pipeline as is used throughout this paper (i.e. fp16, same data and seed) to ensure comparability. With these settings we notice a degradation which is consistent Llama 2's high sensitivity to calibration data we have noticed in prior experiments. When applying our updated search formulation, we observe a major improvement across perplexity and zero-shot results. In particular WikiText2 perplexity reduces from 8.12 to just 6.25. For zero-shot accuracy the increase is even more impressive with an improvement of **9.3 p.p.** over the uniform baseline and strong improvements of 6.0 p.p. over the much more computationally expensive fp32 reference. Interestingly, PTB is the only benchmark where it suffers substantial degradation. However, with the improvements across the board in all other benchmarks, we regard this as an outlier.

In the head-to-head comparison without search, due to the differences between standard SVD and BasisSharing, we compare KFAC-SVD against BasisSharing under both matched memory and matched compute budgets using the same data and

*Table 10.* Llama 2-7B compressed with BasisSharing to 0.8 parameter count with the data and precision used in the original work, the data and precision of our work and with LEMS applied to it.

| Method | Wiki ↓ | C4 ↓ | PTB ↓ | Openb. ↑ | ARC_e ↑ | ARC_c ↑ | WinoG. ↑ | HellaS. ↑ | PIQA ↑ | **Avg. ↑** |
|---|---|---|---|---|---|---|---|---|---|---|
| Original | 5.47 | 7.29 | 24.09 | 0.33 | 0.75 | 0.43 | 0.69 | 0.57 | 0.78 | 59.43 |
| BasisSharing (Ref. data + fp32) | 7.71 | 16.21 | **59.47** | 0.29 | 0.63 | 0.31 | 0.64 | 0.44 | 0.70 | 50.17 |
| BasisSharing (Our data + fp16) | 8.12 | 19.35 | 106.48 | 0.26 | 0.53 | 0.24 | 0.63 | 0.42 | 0.67 | 45.83 |
| +**LEMS** | **6.25** | **9.83** | 297.47 | **0.32** | **0.72** | **0.39** | **0.67** | **0.52** | **0.75** | **56.17** |

*Table 11.* Comparing KFAC-SVD with BasisSharing at matched compute (TFLOPs) and memory (GParams) budgets with respect to the linear layers that are considered in the compression of Llama 2-7B *without any search* applied.

| Method | GParams | TFLOPs | Wiki ↓ | C4 ↓ | PTB ↓ | Openb. ↑ | ARC_e ↑ | ARC_c ↑ | WinoG. ↑ | HellaS. ↑ | PIQA ↑ | **Avg. ↑** |
|---|---|---|---|---|---|---|---|---|---|---|---|---|
| Original | 6.48 | 26.53 | 5.47 | 7.29 | 24.09 | 0.33 | 0.75 | 0.43 | 0.69 | 0.57 | 0.78 | 59.43 |
| Matched **Compute** Budget: 0.8 · Original Linear Layer Compute | | | | | | | | | | | | |
| BasisSharing | 4.53 | **21.40** | 10.16 | 32.60 | 231.67 | 0.25 | 0.51 | 0.24 | 0.62 | 0.36 | 0.63 | 43.50 |
| KFAC-SVD | 5.18 | **21.22** | 7.58 | 15.95 | 70.66 | 0.26 | 0.64 | 0.30 | 0.63 | 0.42 | 0.69 | 49.00 |
| Matched **Memory** Budget: 0.8 · Original Linear Layer Memory | | | | | | | | | | | | |
| BasisSharing | **5.18** | 24.46 | 8.12 | 19.35 | 106.48 | 0.26 | 0.53 | 0.24 | 0.63 | 0.42 | 0.67 | 45.83 |
| KFAC-SVD | **5.18** | 21.22 | 7.58 | 15.95 | 70.66 | 0.26 | 0.64 | 0.30 | 0.63 | 0.42 | 0.69 | 49.00 |

precision. The results of the comparison are presented in Table 11. As can be seen, for both matching memory and matching compute our method comes out ahead in perplexity and zero-shot comparisons, positioning it as a capable alternative to BasisSharing in both scenarios.

## B.6. Calibration Data Ablations

Since our method uses data-based calibration, we investigate the impact the data has on the final result. For that we investigate the impact of the sample count, the sample seed and the dataset.

**Impact of Sample Count.** We start with investigating the sample count. In Figure 11 (left), the performance improves when more samples become available, as one would expected. However, the improvements when going from 64 to 512 (8×) samples are small with perplexity improving from 10.58 to 10.30.

**Impact of Random Seed.** We continue with the evaluation of the random seed (Figure 11, right). Similar to the sample count impact, the seed impact is low, with the largest difference being 0.04 perplexity between seed 42 and seed 2025. This underlines the stability of the presented method.

**Impact of Calibration Dataset.** In all experiments, we used the WikiText2 dataset for calibration. To understand the impact of the calibration set, we repeated the experiments for the 7B and 8B models using the fineweb 16M dataset introduced by (Penedo et al., 2024). This dataset was chosen for its diversity and absence from most common comparisons, allowing for an unbiased evaluation. The results are given in Table 12. When comparing the results to the WikiText2 calibration, it is clear that the calibration set has a significant impact on the performance of all methods. While the WikiText2 performance, as expected, drops significantly, PTB, C4 and the classification task performance improve. Specifically, for our method at 0.7 compression, on Llama 3-8B the accuracy improves from 40.73 to 44.47 without search and from 49.39 to 51.06 with search. Similarly, the performance improves from 45.76 to 46.33 without and from 51.52 to 51.85 with search for Mistral-7B and from 46.78 to 50.04 without and from 53.80 to 55.03 with search for Qwen3-8B, experiencing the biggest improvement. Other methods see comparable improvements e.g. SVD-LLM which improves by 1.07 p.p. and 1.93 p.p. for 0.7 and 0.9 on Qwen3-8B, respectively. These improvements highlight the higher quality of the fineweb dataset compared to its WikiText2 counterpart. Despite the improvements for the baseline methods, our method still outperforms baselines, demonstrating effectiveness independent of the dataset used.

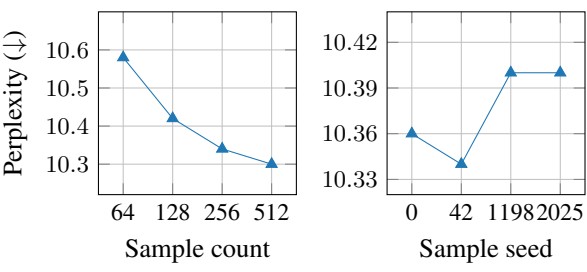

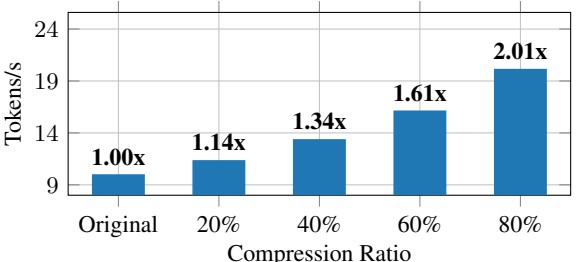

*Figure 11.* Perplexity of Qwen3-8B when compressed using different sample counts and different sampling seeds when removing 20% of parameters with KFAC-SVD + LEMS.

*Figure 12.* Throughput (token/s) of the baseline Qwen3-8B and LEMS compressed variants at different compression ratios.

*Table 12.* Comparison of **KFAC-SVD** with activation- and Fisher-based SVD across modern LLMs under uniform compression and **LEMS** on the **fineweb 16M** dataset.

| | Method | Mistral-7B | | | | Llama 3-8B | | | | Qwen3-8B | | | |
|---|---|---|---|---|---|---|---|---|---|---|---|---|---|
| | | Wiki ↓ | C4 ↓ | PTB ↓ | Acc ↑ | Wiki ↓ | C4 ↓ | PTB ↓ | Acc ↑ | Wiki ↓ | C4 ↓ | PTB ↓ | Acc↑ |
| | Baseline | 5.25 | 8.10 | 27.72 | 63.96 | 6.14 | 9.47 | 9.90 | 63.33 | 9.71 | 15.52 | 15.43 | 62.04 |
| 0.9 | FWSVD | 9.98 | 13.72 | 59.96 | 57.61 | 51.29 | 50.99 | 95.82 | 51.65 | 27.03 | 25.24 | inf | 50.02 |
| | ASVD | 9.49 | 13.22 | 47.53 | 56.73 | 90.72 | 91.79 | 295.15 | 46.77 | 20.52 | 29.17 | 37.16 | 50.90 |
| | SVD-LLM | 8.23 | 11.20 | 51.89 | 57.32 | 21.93 | 19.58 | 36.73 | 54.11 | 16.23 | 19.85 | 22.06 | 57.89 |
| | SVD-LLMv2 | 8.24 | 11.20 | 51.89 | 57.37 | 21.97 | 19.62 | 36.87 | 54.16 | 16.20 | 19.85 | 22.06 | 57.87 |
| | DOBI-SVD | 9.11 | 12.04 | 58.58 | 55.88 | 22.67 | 20.80 | 28.21 | 52.79 | 15.89 | 20.68 | 23.30 | 56.56 |
| | GFWSVD | 19.17 | 26.25 | 124.49 | 54.43 | 1678.71 | 1034.22 | 5110.55 | 37.20 | 101.61 | 80.69 | 214.26 | 50.05 |
| | **KFAC-SVD** | 7.62 | 10.67 | 47.34 | 57.38 | 15.86 | 16.91 | 19.93 | 55.96 | 15.25 | 18.58 | 20.56 | 58.63 |
| | **+LEMS** | **5.57** | **8.37** | **29.40** | **62.84** | **7.02** | **10.69** | **11.76** | **62.58** | **10.10** | **15.86** | **15.80** | **62.56** |
| 0.7 | FWSVD | 40.49 | 45.26 | 228.97 | 44.94 | 1006.33 | 705.28 | 2432.99 | 35.33 | 113.35 | 133.04 | inf | 41.20 |
| | ASVD | 31.60 | 40.65 | 152.53 | 44.87 | 11607.13 | 7095.30 | 25751.55 | 32.94 | 73.18 | 87.59 | 129.95 | 42.77 |
| | SVD-LLM | 29.45 | 21.97 | 138.34 | 45.60 | 181.84 | 85.56 | 391.02 | 41.27 | 44.04 | 30.81 | 69.02 | 46.83 |
| | SVD-LLMv2 | 29.57 | 22.02 | 138.34 | 45.62 | 182.55 | 85.89 | 394.08 | 41.23 | 44.39 | 30.93 | 69.56 | 46.66 |
| | DOBI-SVD | 28.05 | 24.23 | 140.51 | 44.12 | 154.93 | 73.76 | 195.09 | 40.35 | 42.85 | 31.11 | 66.63 | 48.14 |
| | GFWSVD | 941.67 | 805.45 | 2180.92 | 36.87 | 26987.39 | 9473.46 | 23264.58 | 32.14 | 12164.17 | 3248.46 | 7095.30 | 35.17 |
| | **KFAC-SVD** | 23.44 | 20.05 | 95.82 | 46.33 | 80.06 | 47.34 | 134.60 | 44.47 | 38.87 | 28.05 | 53.44 | 50.04 |
| | **+LEMS** | **12.50** | **13.33** | **57.67** | **51.85** | **21.55** | **21.21** | **38.41** | **51.06** | **16.72** | **20.28** | **24.47** | **55.03** |

## B.7. Inference Acceleration

To quantify the practical efficiency gains of SVD when used with LEMS, we evaluated the wall-clock inference speedup of the compressed Qwen3-8B model against the uncompressed baseline. All measurements were conducted on a single NVIDIA A100 GPU. Adopting the evaluation protocol from prior work (Wang et al., 2025b), we fixed the batch size to 4, with a prefill sequence length of 1024 tokens and a decoding length of 256 tokens. As illustrated in Figure 12, LEMS demonstrates a consistent improvement in generation speed as the model is compressed further. The method achieves a $1.14\times$ speedup at a 20% compression ratio, scaling up to $1.61\times$ at 60% compression. At the most aggressive setting of 80% compression, the model reaches a substantial $2.01\times$ speedup over the dense baseline.

## B.8. Qualitative Results

We present qualitative generations from SVD-compressed Qwen3-8B models *without* any post-compression finetuning. We compare KFAC-SVD and LEMS against SVD-LLM, a state-of-the-art SVD-based compression baseline. Table 13 reports representative model outputs for the same prompt across compression ratios, illustrating the effects of compression on response coherence, formatting, and content fidelity.

*Table 13.* Example content generated as response to "What is the responsibility of the AI assistant?" by Qwen3-8B models that were compressed with SVD-LLM and KFAC-SVD without any finetuning under different compression ratios.

| Ratio | KFAC-SVD | SVD-LLM |
|---|---|---|
| Original | **What is the responsibility of the AI assistant?** And how can it be ensured? The responsibility and accountability of AI assistants are critical issues that require careful consideration. Here's a structured analysis of the key points and a comprehensive approach to ensuring responsibility: 
 1. **Ethical and Moral Responsibility**: AI systems should be developed and used in a way that aligns with ethical and moral principles, avoiding harm to individuals and society. 2. ... | |
| 10% | **What is the responsibility of the AI assistant?** And how can it be ensured? The responsibility (ethical, legal, technical, etc.) of AI assistants is a critical area of concern in the development and application of artificial intelligence technology. This issue is not only related to the fairness and impartiality of the AI system itself but also involves the rights and interests of individuals and society. In order to ensure the correct development of intelligent assistants, we need to establish a comprehensive framework that involves both technical and non-technical dimensions. ... | **What is the responsibility of the AI assistant?** - - - According to its creators, AI assistants are designed to be helpful and to support users. - = - They should not act as a replacement for human intelligence and should instead focus on assisting users through tasks they can perform. = = - - - - However, some AI systems have been designed with specific purposes, such as responding to questions and carrying out tasks that require human thinking. For example, there are AI programs that are used in hospitals that perform complex tasks such as interpreting X-rays and diagnosing diseases. ... |
| 20% | **What is the responsibility of the AI assistant?** According to the given prompt, the assistant is supposed to be a helpful companion who provides the best responses. It is not supposed be something of the dark side. So the AI should not be malevolent, but rather a positive and helpful entity. But, in certain situations, even with all the good intentions, perhaps the Dark Side could come into play. Also, there are some guidelines. For example, if there is a request for harm, then the system should refuse. The AI must not provide harmful information. Therefore, to avoid any harm and to maintain a beneficial environment, it is important to ensure that the responses are safe and positive. ... | **What is the responsibility of the AI assistant?** What is the responsibility of an AI assistant? and what should they do to be ethical? 
 5.00 PM, 15 March 2021 
 Please help me to create a presentation about AI assistants in the modern age. 
 Please share your views and opinions about what an ethical AI should be like in today's world. 
 I want to present about the AI as assistants to humans and how they are being developed. I am in charge of a department to which AI is being introduced for the first time. ... |
| 30% | **What is the responsibility of the AI assistant?** And the main point is to be on time and precise. 
 Which of the four are the most important. Which is most beneficial? Which one is a good solution for the whole team. I want to do the assignment for each one, but I need to clarify it. I'm not the best at it, and need help from someone. So I will have to provide some clarification. To get the support of someone who is willing to help, I have the authority. It is not clear, or maybe I don't know about the question and the meaning. ... | **What is the responsibility of the AI assistant?** and it will give the answer. So to answer questions, the AI may be asked the question by someone, and perhaps I know something about the subject, or perhaps he can be helpful in this situation. 
 So the person can explain to the audience that he wants to be the assistant, because of his own identity. So he has been given to an assistant by the creator and can now become an employee. This idea has to exist in a situation, but the human wants the job. He is responsible for maintaining a position to create AI. |

## C. Algorithms and Implementation Details

**End-to-End Compression Pipeline.**   Algorithm 1 outlines the high-level workflow of our framework. Crucially, the pipeline is modular, decoupling the local decomposition strategy from the global rank allocation.

1. **Phase I: Decomposition.** We iterate through layers to compute decompositions, including the singular value spectra. While this work proposes the robust KFAC-SVD estimator, this step can be substituted with any SVD variant (e.g., standard SVD, activation-SVD) provided it yields a valid singular value spectrum for the search.

2. **Phase II: Global Rank Allocation (LEMS).** We solve for the optimal integer ranks $\{r_\ell^\star\}$ that maximize performance under the parameter budget $P_{\text{target}}$. This search utilizes the precomputed spectra but is agnostic to how they were generated.

3. **Phase III: Reconstruction.** Layers are replaced with their low-rank factorizations. For data dependent methods, this may optionally involve recalibrating statistics used in the decomposition, but is not used for our results.

---

**Algorithm 1** High-Level Procedure: Modular SVD Compression with Global Search

---

1: **Input:** pretrained model $M$; calibration sets $\mathcal{C}$ (general), $\mathcal{C}_S$ (sensitivity); budget $P_{\text{target}}$
2: **Input:** candidate ranks $\{\mathcal{R}_\ell\}$; flag `recalibrate` $\in \{0,1\}$
3: **Output:** compressed model $M'$
4: **procedure** COMPRESSMODEL($M, \mathcal{C}, \mathcal{C}_S, P_{\text{target}}, \{\mathcal{R}_\ell\}$, `recalibrate`)
    *# (I) Decomposition Phase (Modular)*
5:   **for** each compressible linear layer $\ell$ in $M$ **do**
6:     {Compute SVD factors (e.g., via Alg. 2 or other SVD approaches)}
7:     $\mathcal{D}_\ell \leftarrow$ DECOMPOSELAYER($M, \mathcal{C}_S, \ell$)
8:   **end for**
    *# (II) Global Rank Allocation (LEMS)*
9:   {Solve ILP using Shape/Scale/Bias error model (Alg. 3)}
10:   $\{r_\ell^\star\}_\ell \leftarrow$ LEMS_SEARCH($M, \{\mathcal{R}_\ell\}, \{\mathcal{D}_\ell\}_\ell, P_{\text{target}}, \mathcal{C}$)
    *# (III) Reconstruction Phase*
11:   $M' \leftarrow M$
12:   **for** each compressible layer $\ell$ in forward order **do**
13:     **if** `recalibrate` $= 1$ **then**
14:       $\mathcal{D}_\ell \leftarrow$ DECOMPOSELAYER($M', \mathcal{C}, \ell$) {Optional: Recalculate on updated $M'$}
15:     **end if**
16:     retrieve $\mathbf{U}_\ell, \mathbf{\Sigma}_\ell, \mathbf{V}_\ell, \mathbf{L}_{A,\ell}, \mathbf{L}_{G,\ell}$ from $\mathcal{D}_\ell$
17:     {Reconstruct low-rank weights (example for KFAC-SVD, for Standard SVD, $\mathbf{L} = \mathbf{I}$)}
18:     $\mathbf{C}_\ell \leftarrow \mathbf{L}_{G,\ell}^{-\top} \mathbf{U}_{r_\ell^\star} \mathbf{\Sigma}_{r_\ell^\star}^{1/2}$
19:     $\mathbf{B}_\ell \leftarrow \mathbf{\Sigma}_{r_\ell^\star}^{1/2} \mathbf{V}_{r_\ell^\star}^\top \mathbf{L}_{A,\ell}^{-1}$
20:     replace $\mathbf{W}_\ell$ in $M'$ with $\mathbf{C}_\ell \mathbf{B}_\ell$
21:   **end for**
22:   **return** $M'$
23: **end procedure**

---

**KFAC Fisher-Weighted SVD Decomposition.** The core of our SVD method is the robust estimation of the whitening matrices, detailed in Algorithm 2. Unlike prior Fisher-based SVD (Chekalina et al., 2025) that suffered from rank deficiency ($k \ll d_{\text{in}}d_{\text{out}}$), we use a decoupled token-wise estimation (Lines 4-8). To ensure accurate scaling of the covariance matrices, $\widehat{\mathbf{A}}$ and $\widehat{\mathbf{G}}$, we explicitly count only non-masked tokens for the normalization factor $N$ (Line 7), preventing the underestimation of curvature in batches with significant padding. To ensure numerical stability, we process these raw estimates using MAKESPD (Line 10). This routine symmetrizes the matrix $\mathbf{M}$ and applies a minimal dynamic diagonal shift to ensure the condition $\lambda_{\min}(\mathbf{M}) \geq \tau \cdot \text{mean}(|\text{diag}(\mathbf{M})|)$ is met, guaranteeing that the subsequent Cholesky decomposition succeeds.

Implementation Note on Whitening: While the main text formulates the whitening transform using the symmetric square root $\mathbf{G}^{1/2}$ for notational clarity, our implementation uses Cholesky factors for numerical efficiency. Specifically, we compute $\mathbf{G} = \mathbf{L}_G \mathbf{L}_G^\top$ and form the whitened weights as $\widetilde{\mathbf{W}} = \mathbf{L}_G^\top \mathbf{W} \mathbf{L}_A$ (Line 12). This transformation is isometric to the symmetric square root formulation with respect to the Frobenius norm objective, yielding identical singular values and optimal low-rank indices.

---

**Algorithm 2** KFAC_SVD($M, \mathcal{D}_{cal}, \ell$): Estimate Factors and Compute Whitened SVD

---

1: **Input:** model $M$; data $\mathcal{D}_{cal}$; target layer $\ell$ with weights $\mathbf{W}_\ell$
2: **Output:** decomposition cache $\mathcal{D}_\ell = \{\mathbf{U}_\ell, \boldsymbol{\Sigma}_\ell, \mathbf{V}_\ell, \mathbf{L}_{A,\ell}, \mathbf{L}_{G,\ell}\}$
    *# (I) Streaming Curvature Estimation*
3: initialize $\mathbf{S}_A \leftarrow \mathbf{0}, \mathbf{S}_G \leftarrow \mathbf{0}, N \leftarrow 0$
4: **for** mini-batch $(x, \text{mask}) \sim \mathcal{D}_{cal}$ **do**
5:     Forward/Backward pass: Extract $X_\ell$ (stacked inputs $x$) and $\Delta_\ell$ (stacked grads $\delta$)
6:     $\mathbf{S}_A \leftarrow \mathbf{S}_A + X_\ell^\top X_\ell; \quad \mathbf{S}_G \leftarrow \mathbf{S}_G + \Delta_\ell^\top \Delta_\ell$
7:     $N \leftarrow N + \text{COUNT}(mask)$ {Token-wise accumulation}
8: **end for**
9: $\widehat{\mathbf{A}}_\ell \leftarrow \frac{1}{N}\mathbf{S}_A; \quad \widehat{\mathbf{G}}_\ell \leftarrow \frac{1}{N}\mathbf{S}_G$
10: $\mathbf{A}_\ell, \mathbf{G}_\ell \leftarrow \text{MAKESPD}(\widehat{\mathbf{A}}_\ell), \text{MAKESPD}(\widehat{\mathbf{G}}_\ell)$ {Normalize & damp}
    *# (II) Whitening and Decomposition*
11: Compute Cholesky: $\mathbf{A}_\ell = \mathbf{L}_{A,\ell}\mathbf{L}_{A,\ell}^\top, \mathbf{G}_\ell = \mathbf{L}_{G,\ell}\mathbf{L}_{G,\ell}^\top$
12: $\widetilde{\mathbf{W}}_\ell \leftarrow \mathbf{L}_{G,\ell}^\top \mathbf{W}_\ell \mathbf{L}_{A,\ell}$ {Project to whitened basis}
13: $(\mathbf{U}_\ell, \boldsymbol{\Sigma}_\ell, \mathbf{V}_\ell) \leftarrow \text{SVD}(\widetilde{\mathbf{W}}_\ell)$
14: **return** $\mathcal{D}_\ell \leftarrow \{\mathbf{U}_\ell, \boldsymbol{\Sigma}_\ell, \mathbf{V}_\ell, \mathbf{L}_{A,\ell}, \mathbf{L}_{G,\ell}\}$

**Global Rank Allocation.** Algorithm 3 details the LEMS procedure. This process converts the raw Fisher spectral energy into a calibrated additive proxy as discussed in Section 4.2. The procedure begins by anchoring the **scale** of the error: for each layer $\ell$, we measure the empirical KL-divergence at a reference compression (e.g., $c_{\text{ref}} = 10\%$) to determine the scalar importance $\alpha_\ell$ (Line 6). Next, we address the error propagation correction using our **bias**-function. We parameterize the bias $\beta(d_\ell)$ using a harmonic decay function controlled by the block based depth $d_\ell$ of layer $\ell$, magnitude $\kappa$ and decay rate $\gamma$ (Eq. 17). This formulation assigns higher penalties to earlier blocks to counteract the *error explosion* effect (Huang et al., 2025). Crucially, because the optimal propagation penalty depends on the specific architecture (depth, residual connections), we cannot use fixed values. Instead, we optimize $\kappa$ and $\gamma$ via a bilevel optimization loop (Lines 8-25). We employ a TPE sampler provided by (Akiba et al., 2019) to propose candidate $(\kappa, \gamma)$ pairs. For each pair, we: 1. Construct Costs: Calculate the layer-wise costs $L_\ell(r)$ using the current bias parameters (Lines 11-16). 2. Solve ILP: Obtain the optimal rank configuration for this specific cost landscape (Line 18). 3. Validate: Temporarily apply these ranks and measure the true validation loss (Line 19, 20). This *loop-in-the-loop* ensures that the error model is dynamically fitted to maximize the generalization of the final rank assignment.

---

**Algorithm 3** LEMS_SEARCH(...): Bilevel Optimization for Rank Allocation

---

1: **Input:** caches $\{\mathcal{D}_\ell\}$; budget $P_{\text{target}}$; calibration $\mathcal{C}$; search trials $N_{\text{trials}}$
2: **Output:** optimal ranks $\{r_\ell^\star\}$
    *# (I) Scale Calibration (Scale $\alpha_\ell$)*
3: Precompute uncompressed outputs $Y_{\text{orig}}$ on $\mathcal{C}$
4: **for** each layer $\ell \in \{1 \ldots L\}$ **do**
5:     Measure KL-div $k_\ell$ at reference rank $r_{\text{ref}}$ (e.g. 10% param)
6:     $\alpha_\ell \leftarrow k_\ell / (S_\ell(r_{\text{ref}}) + \epsilon)$ {Eq. (4)}
7: **end for**
    *# (II) Bilevel Search (Bias $\kappa, \gamma$)*
8: Initialize TPE sampler $\mathcal{S}$;    $\mathcal{H}^* \leftarrow \infty$
9: **for** $k = 1$ to $N_{\text{trials}}$ **do**
10:     Sample bias params $\kappa_k, \gamma_k \leftarrow \mathcal{S}.\text{suggest}(\ldots)$
    *# (A) Construct Calibrated Proxy*
11:     $H_{\text{norm}} \leftarrow \sum_{j=1}^{B} j^{-\gamma_k}$
12:     **for** each layer $\ell$ **do**
13:         $\beta(d_\ell) \leftarrow \kappa_k \cdot (\sum_{j=d_\ell}^{B} j^{-\gamma_k}) / H_{\text{norm}}$ {Eq. (6)}
14:         **for** each candidate $r \in \mathcal{R}_\ell$ **do**
15:             $\text{Cost}_{\ell, r} \leftarrow \underbrace{S_\ell(r)}_{\text{Shape}} \cdot \underbrace{\alpha_\ell}_{\text{Scale}} \cdot \underbrace{(1 + \beta(d_\ell))}_{\text{Bias}}$
16:         **end for**
17:     **end for**
    *# (B) Inner Loop: Solve ILP*
18:     $\{r_\ell^{(k)}\} \leftarrow \text{SOLVEILP}(\{\text{Cost}_{\ell, r}\}, P_{\text{target}})$
    *# (C) Outer Loop: Validate*
19:     Apply ranks $\{r_\ell^{(k)}\}$ to model $M_{\text{tmp}}$
20:     $\mathcal{L}_{\text{val}} \leftarrow \text{EVALUATEKL}(M_{\text{tmp}}, Y_{\text{orig}}, \mathcal{C})$
21:     $\mathcal{S}.\text{observe}(\kappa_k, \gamma_k, \mathcal{L}_{\text{val}})$
22:     **if** $\mathcal{L}_{\text{val}} < \mathcal{H}^*$ **then**
23:         $\mathcal{H}^* \leftarrow \mathcal{L}_{\text{val}}$;    $\{r_\ell^\star\} \leftarrow \{r_\ell^{(k)}\}$
24:     **end if**
25: **end for**
26: **return** $\{r_\ell^\star\}$

---

# D. Detailed Result Tables

**Detailed Search Table.** Table 1 in the main content only reports the average across the zero-shot datasets. Here we present the full results, with all individual accuracy values. The detailed results for 0.8 ratio are presented in Table 16. The results for 0.6 are presented in Table 17.

**Detailed SVD Table.** In the main content Table 2 only reports the average across the zero-shot datasets. In Table 14 and Table 15 we present the detailed results with all individual accuracy values for 0.9 and 0.7 compression ratio, respectively.

**Numerical Results for Figure 2.** In Table 18, we provide the extended version of Figure 2, which includes the detailed data for all models. Please note that because of stability issues for SVD-LLM on the 32B model, we opted for SVD-LLMv2 instead, which performs nearly identical in our experiments but is much more expensive due to its SVD-based whitening.

*Table 14.* Extended format zero-shot performance and perplexity of LLMs (Mistral-7B, Llama 3-8B, Qwen3-8B) with a **0.9 parameter retention** compressed with a range of **SVD baselines** and KFAC-SVD. **Bold**/Underline denote best/second-best compressed results.

| | Method | Wiki ↓ | C4 ↓ | PTB ↓ | Openb. ↑ | ARC_e ↑ | ARC_c ↑ | WinoG. ↑ | HellaS. ↑ | PIQA ↑ | **Avg.** ↑ |
|---|---|---|---|---|---|---|---|---|---|---|---|
| **Mistral-7B** | Original | 5.25 | 8.10 | 27.72 | 0.35 | 0.80 | 0.51 | 0.75 | 0.62 | 0.81 | 63.96 |
| | FWSVD | 9.47 | 13.75 | 58.35 | 0.31 | 0.74 | 0.40 | 0.69 | 0.51 | 0.76 | 56.93 |
| | ASVD | 9.14 | 13.41 | 49.13 | 0.33 | 0.73 | 0.41 | 0.69 | 0.52 | 0.77 | 57.30 |
| | SVD-LLM | 6.46 | 15.07 | 66.12 | 0.30 | 0.75 | 0.42 | 0.71 | 0.49 | 0.75 | 56.81 |
| | SVD-LLMv2 | 6.46 | 15.07 | 66.12 | 0.30 | 0.75 | 0.42 | 0.71 | 0.49 | 0.75 | 56.78 |
| | DOBI-SVD | 7.11 | 15.19 | 63.58 | 0.29 | 0.71 | 0.38 | 0.71 | 0.49 | 0.74 | 55.17 |
| | GFWSVD | 31.66 | 65.09 | 305.72 | 0.18 | 0.46 | 0.24 | 0.54 | 0.38 | 0.63 | 40.73 |
| | **KFAC-SVD** | 6.22 | 13.10 | 44.04 | 0.29 | 0.75 | 0.42 | 0.70 | 0.50 | 0.75 | 56.92 |
| | +LEMS | **5.37** | **8.49** | **29.86** | **0.34** | **0.79** | **0.49** | **0.74** | **0.59** | **0.80** | **62.77** |
| **Llama 3-8B** | Original | 6.14 | 9.47 | 9.90 | 0.34 | 0.81 | 0.52 | 0.74 | 0.61 | 0.79 | 63.33 |
| | FWSVD | 42.11 | 59.96 | 69.83 | 0.25 | 0.62 | 0.31 | 0.64 | 0.41 | 0.70 | 48.88 |
| | ASVD | 65.09 | 77.60 | 207.67 | 0.27 | 0.57 | 0.31 | 0.63 | 0.41 | 0.68 | 47.89 |
| | SVD-LLM | 10.14 | 39.63 | 49.91 | 0.26 | 0.69 | 0.37 | 0.66 | 0.43 | 0.72 | 52.27 |
| | SVD-LLMv2 | 10.18 | 39.71 | 50.40 | 0.26 | 0.69 | 0.37 | 0.66 | 0.43 | 0.72 | 52.17 |
| | DOBI-SVD | 11.22 | 34.77 | 42.36 | 0.27 | 0.68 | 0.37 | 0.68 | 0.46 | 0.72 | 52.98 |
| | GFWSVD | 2569.75 | 3827.63 | 8491.96 | 0.14 | 0.30 | 0.20 | 0.53 | 0.28 | 0.55 | 33.40 |
| | **KFAC-SVD** | 8.84 | 25.69 | 32.60 | 0.29 | 0.71 | 0.37 | 0.69 | 0.47 | 0.74 | 54.45 |
| | +LEMS | **6.58** | **11.11** | **11.29** | **0.34** | **0.80** | **0.50** | **0.73** | **0.58** | **0.78** | **61.99** |
| **Qwen3-8B** | Original | 9.71 | 15.52 | 15.43 | 0.31 | 0.84 | 0.56 | 0.68 | 0.57 | 0.77 | 62.04 |
| | FWSVD | 16.95 | 24.75 | 31.54 | 0.29 | 0.73 | 0.40 | 0.66 | 0.43 | 0.71 | 53.82 |
| | ASVD | 20.36 | 29.98 | 36.37 | 0.27 | 0.61 | 0.37 | 0.63 | 0.48 | 0.71 | 51.19 |
| | SVD-LLM | 12.52 | 27.40 | 36.66 | 0.31 | 0.71 | 0.44 | 0.66 | 0.50 | 0.73 | 55.96 |
| | SVD-LLMv2 | 12.52 | 27.45 | 36.80 | 0.31 | 0.71 | 0.44 | 0.66 | 0.50 | 0.73 | 55.80 |
| | DOBI-SVD | 13.46 | 28.88 | 35.81 | 0.29 | 0.69 | 0.45 | 0.68 | 0.50 | 0.71 | 55.48 |
| | GFWSVD | 171.49 | 350.51 | 765.57 | 0.27 | 0.50 | 0.34 | 0.58 | 0.43 | 0.67 | 46.66 |
| | **KFAC-SVD** | 11.51 | 23.76 | 28.00 | 0.32 | 0.76 | 0.47 | 0.66 | 0.50 | 0.74 | 57.07 |
| | +LEMS | **9.85** | **16.04** | **15.77** | **0.33** | **0.84** | **0.58** | **0.70** | **0.56** | **0.77** | **62.89** |

*Table 15.* Extended format zero-shot performance and perplexity of LLMs (Mistral-7B, Llama 3-8B, Qwen3-8B) with a **0.7 parameter retention** compressed with a range of **SVD baselines** and KFAC-SVD. **Bold**/Underline denote best/second-best compressed results.

| | Method | Wiki ↓ | C4 ↓ | PTB ↓ | Openb. ↑ | ARC_e ↑ | ARC_c ↑ | WinoG. ↑ | HellaS. ↑ | PIQA ↑ | **Avg.** ↑ |
|---|---|---|---|---|---|---|---|---|---|---|---|
| **Mistral-7B** | Original | 5.25 | 8.10 | 27.72 | 0.35 | 0.80 | 0.51 | 0.75 | 0.62 | 0.81 | 63.96 |
| | FWSVD | 34.84 | 50.30 | 215.10 | 0.21 | 0.59 | 0.25 | 0.60 | 0.34 | 0.66 | 44.24 |
| | ASVD | 28.16 | 42.94 | 163.64 | 0.24 | 0.57 | 0.29 | 0.62 | 0.38 | 0.66 | 46.09 |
| | SVD-LLM | 10.96 | 42.60 | 169.49 | 0.24 | 0.56 | 0.25 | 0.62 | 0.35 | 0.64 | 44.39 |
| | SVD-LLMv2 | 10.96 | 42.60 | 169.49 | 0.24 | 0.56 | 0.25 | 0.62 | 0.35 | 0.64 | 44.35 |
| | DOBI-SVD | 12.37 | 39.71 | 170.82 | 0.22 | 0.50 | 0.26 | 0.60 | 0.35 | 0.65 | 42.94 |
| | GFWSVD | 2549.75 | 3404.36 | 5355.81 | 0.14 | 0.26 | 0.24 | 0.50 | 0.26 | 0.54 | 32.29 |
| | **KFAC-SVD** | 9.29 | 34.64 | 110.72 | 0.22 | 0.61 | 0.28 | 0.62 | 0.36 | 0.66 | 45.76 |
| | **+LEMS** | **7.42** | **18.25** | **67.95** | **0.29** | **0.69** | **0.33** | **0.65** | **0.42** | **0.72** | **51.52** |
| **Llama 3-8B** | Original | 6.14 | 9.47 | 9.90 | 0.34 | 0.81 | 0.52 | 0.74 | 0.61 | 0.79 | 63.33 |
| | FWSVD | 716.39 | 837.54 | 1939.75 | 0.13 | 0.33 | 0.21 | 0.52 | 0.27 | 0.55 | 33.40 |
| | ASVD | 10989.41 | 9399.74 | 36032.89 | 0.14 | 0.30 | 0.20 | 0.50 | 0.27 | 0.56 | 33.03 |
| | SVD-LLM | 34.64 | 376.04 | 600.91 | 0.20 | 0.40 | 0.21 | 0.58 | 0.29 | 0.58 | 37.69 |
| | SVD-LLMv2 | 34.98 | 378.99 | 615.16 | 0.20 | 0.40 | 0.21 | 0.58 | 0.29 | 0.58 | 37.70 |
| | DOBI-SVD | 31.54 | 500.12 | 223.67 | 0.19 | 0.39 | 0.23 | 0.59 | 0.31 | 0.60 | 38.48 |
| | GFWSVD | 41798.89 | 53670.83 | 72222.00 | 0.14 | 0.28 | 0.21 | 0.49 | 0.26 | 0.53 | 31.77 |
| | **KFAC-SVD** | 19.43 | 151.34 | 253.45 | 0.21 | 0.48 | 0.24 | 0.60 | 0.31 | 0.61 | 40.73 |
| | **+LEMS** | **11.07** | **33.51** | **65.09** | **0.26** | **0.63** | **0.33** | **0.65** | **0.40** | **0.69** | **49.39** |
| **Qwen3-8B** | Original | 9.71 | 15.52 | 15.43 | 0.31 | 0.84 | 0.56 | 0.68 | 0.57 | 0.77 | 62.04 |
| | FWSVD | 41.37 | 68.75 | 90.72 | 0.20 | 0.58 | 0.27 | 0.56 | 0.33 | 0.63 | 42.68 |
| | ASVD | 70.66 | 96.95 | 131.49 | 0.24 | 0.49 | 0.29 | 0.55 | 0.38 | 0.65 | 43.10 |
| | SVD-LLM | 17.11 | 63.09 | 136.19 | 0.24 | 0.60 | 0.30 | 0.59 | 0.38 | 0.65 | 45.76 |
| | SVD-LLMv2 | 17.15 | 63.33 | 136.72 | 0.23 | 0.60 | 0.30 | 0.58 | 0.38 | 0.65 | 45.62 |
| | DOBI-SVD | 18.54 | 63.83 | 94.34 | 0.25 | 0.55 | 0.32 | 0.63 | 0.39 | 0.64 | 46.24 |
| | GFWSVD | 21684.98 | 27627.38 | 50814.52 | 0.17 | 0.31 | 0.22 | 0.51 | 0.28 | 0.56 | 34.06 |
| | **KFAC-SVD** | 14.69 | 50.00 | 93.97 | 0.25 | 0.58 | 0.32 | 0.62 | 0.38 | 0.64 | 46.78 |
| | **+LEMS** | **11.81** | **28.33** | **37.97** | **0.29** | **0.74** | **0.41** | **0.64** | **0.43** | **0.72** | **53.80** |

*Table 16.* Extended format zero-shot performance and perplexity of LLMs (Mistral-7B, Llama 3-8B, Qwen3-8B) compressed with a range of **different searches** with a **0.8 parameter retention**. **Bold**/Underline denote best/second-best compressed results.

| | Search Method | Wiki ↓ | C4 ↓ | PTB ↓ | Openb. ↑ | ARC_e ↑ | ARC_c ↑ | WinoG. ↑ | HellaS. ↑ | PIQA ↑ | **Avg.** ↑ |
|---|---|---|---|---|---|---|---|---|---|---|---|
| **Mistral-7B** | uniform | 7.14 | 19.43 | 64.84 | 0.26 | 0.70 | 0.36 | **0.69** | 0.43 | 0.70 | 52.35 |
| | ASVD | 7.20 | 22.41 | 78.51 | 0.25 | 0.63 | 0.29 | 0.64 | 0.38 | 0.68 | 47.81 |
| | SVD-LLMv2 | 7.13 | 19.35 | 64.84 | 0.27 | 0.70 | 0.37 | **0.69** | 0.43 | 0.70 | 52.68 |
| | MRCS | 7.11 | 19.13 | 68.21 | 0.24 | 0.70 | 0.35 | 0.68 | 0.43 | 0.70 | 51.75 |
| | ARS | 7.26 | 11.74 | 47.71 | 0.28 | 0.71 | 0.38 | 0.68 | 0.48 | 0.75 | 54.58 |
| | ATP | 7.14 | 19.43 | 64.84 | 0.26 | 0.70 | 0.36 | **0.69** | 0.43 | 0.70 | 52.35 |
| | **LEMS** | **5.98** | **11.24** | **37.09** | **0.34** | **0.76** | **0.41** | **0.69** | **0.50** | **0.77** | **57.67** |
| **Llama 3-8B** | uniform | 11.44 | 52.00 | 80.06 | 0.24 | 0.61 | 0.31 | 0.64 | 0.38 | 0.68 | 47.69 |
| | ASVD | 12.92 | 63.83 | 99.64 | 0.25 | 0.60 | 0.27 | 0.62 | 0.35 | 0.67 | 45.95 |
| | SVD-LLMv2 | 11.40 | 52.00 | 80.38 | 0.24 | 0.62 | 0.30 | 0.64 | 0.38 | 0.68 | 47.70 |
| | MRCS | 11.99 | 55.89 | 83.91 | 0.24 | 0.59 | 0.29 | 0.63 | 0.37 | 0.68 | 46.63 |
| | ARS | 11.81 | 20.44 | 28.83 | 0.28 | 0.71 | **0.39** | 0.68 | 0.47 | 0.72 | 54.31 |
| | ATP | 11.07 | 35.32 | 81.32 | 0.26 | 0.67 | 0.36 | 0.67 | 0.43 | 0.70 | 51.56 |
| | **LEMS** | **8.16** | **17.52** | **22.80** | **0.32** | **0.74** | **0.39** | **0.69** | **0.48** | **0.75** | **55.99** |
| **Qwen3-8B** | uniform | 12.52 | 31.85 | 47.25 | **0.30** | 0.71 | 0.42 | 0.65 | 0.44 | 0.69 | 53.56 |
| | ASVD | 15.70 | 43.79 | 72.61 | 0.21 | 0.65 | 0.33 | 0.62 | 0.38 | 0.68 | 47.72 |
| | SVD-LLMv2 | 12.52 | 31.72 | 47.44 | **0.30** | 0.70 | 0.42 | 0.64 | 0.44 | 0.69 | 53.36 |
| | MRCS | 14.52 | 37.38 | 53.75 | 0.29 | 0.66 | 0.39 | 0.65 | 0.44 | 0.68 | 52.04 |
| | ARS | 11.58 | 22.41 | 25.05 | 0.27 | 0.77 | 0.46 | 0.65 | 0.45 | 0.74 | 55.66 |
| | ATP | 12.52 | 31.85 | 47.25 | **0.30** | 0.71 | 0.42 | 0.65 | 0.44 | 0.69 | 53.56 |
| | **LEMS** | **10.38** | **18.32** | **18.32** | **0.30** | **0.80** | **0.51** | **0.68** | **0.51** | **0.76** | **59.28** |

*Table 17.* Extended format zero-shot performance and perplexity of LLMs (-7B, Llama 3-8B, -8B) compressed with a range of **different searches** with a **0.6 parameter retention**. **Bold**/Underline denote best/second-best compressed results.

|  | Search Method | Wiki↓ | C4↓ | PTB↓ | Openb.↑ | ARC_e↑ | ARC_c↑ | WinoG.↑ | HellaS.↑ | PIQA↑ | **Avg.↑** |
|---|---|---|---|---|---|---|---|---|---|---|---|
| **Mistral-7B** | uniform | 14.38 | 72.05 | 208.48 | 0.19 | 0.47 | 0.21 | 0.57 | 0.32 | 0.59 | 39.10 |
|  | ASVD | 16.78 | 102.00 | 317.89 | 0.16 | 0.38 | 0.20 | 0.56 | 0.29 | 0.56 | 35.56 |
|  | SVD-LLMv2 | 14.90 | 73.18 | 209.30 | 0.19 | 0.47 | 0.21 | 0.58 | 0.32 | 0.59 | 39.29 |
|  | MRCS | 14.21 | 74.05 | 253.45 | 0.19 | 0.43 | 0.19 | 0.55 | 0.31 | 0.57 | 37.44 |
|  | ARS | 19.43 | 37.09 | 166.87 | 0.19 | 0.49 | 0.23 | 0.57 | 0.34 | 0.64 | 40.83 |
|  | ATP | 13.59 | 60.43 | 189.08 | 0.19 | 0.48 | 0.24 | 0.58 | 0.33 | 0.60 | 40.24 |
|  | **LEMS** | **10.58** | **34.37** | **124.00** | **0.22** | **0.59** | **0.28** | **0.62** | **0.36** | **0.66** | **45.60** |
| **Llama 3-8B** | uniform | 48.56 | 475.36 | 1312.50 | 0.16 | 0.33 | 0.20 | 0.54 | 0.28 | 0.55 | 34.39 |
|  | ASVD | 75.21 | 571.15 | 1577.00 | 0.16 | 0.33 | 0.19 | 0.53 | 0.28 | 0.55 | 33.98 |
|  | SVD-LLMv2 | 47.53 | 467.99 | 1242.65 | 0.16 | 0.33 | 0.19 | 0.54 | 0.28 | 0.56 | 34.43 |
|  | MRCS | 67.95 | 610.37 | 1815.12 | 0.15 | 0.32 | 0.20 | 0.53 | 0.27 | 0.54 | 33.41 |
|  | ARS | 28.77 | **56.55** | **112.03** | 0.19 | 0.50 | 0.22 | 0.55 | 0.33 | **0.64** | 40.57 |
|  | ATP | 25.14 | 173.51 | 467.99 | 0.18 | 0.43 | 0.23 | **0.62** | 0.32 | 0.62 | 39.95 |
|  | **LEMS** | **17.86** | 81.01 | 201.28 | **0.21** | **0.51** | **0.26** | **0.62** | **0.35** | **0.64** | **43.09** |
| **Qwen3-8B** | uniform | 21.68 | 101.61 | 195.85 | 0.19 | 0.43 | 0.24 | 0.59 | 0.33 | 0.61 | 39.63 |
|  | ASVD | 29.22 | 122.56 | 228.97 | 0.15 | 0.38 | 0.20 | 0.54 | 0.30 | 0.60 | 36.09 |
|  | SVD-LLMv2 | 21.42 | 96.95 | 189.82 | 0.19 | 0.43 | 0.23 | 0.59 | 0.33 | 0.61 | 39.74 |
|  | MRCS | 38.72 | 161.10 | 322.90 | 0.16 | 0.38 | 0.20 | 0.56 | 0.30 | 0.58 | 36.42 |
|  | ARS | 29.51 | 83.91 | 161.10 | 0.18 | 0.50 | 0.23 | 0.58 | 0.32 | 0.63 | 40.72 |
|  | ATP | 19.47 | 84.89 | 186.88 | 0.19 | 0.43 | 0.25 | 0.59 | 0.34 | 0.61 | 40.20 |
|  | **LEMS** | **15.70** | **57.22** | **120.66** | **0.23** | **0.60** | **0.29** | **0.60** | **0.35** | **0.66** | **45.51** |

*Table 18.* Detailed performance evaluation of **LEMS** for both KFAC-SVD and SVD-LLM across five Qwen3 model sizes (1.7B-32B). The table contrasts the baseline uniform rank allocation with our proposed LEMS across compression ratios $c \in [0.5, 0.9]$. **Bold** indicates the best result between the uniform and search-based allocation for each method, showing the massive improvements brought about by LEMS.

| | Method | Qwen3 | | | | | | | | | |
|---|---|---|---|---|---|---|---|---|---|---|---|
| | | 1.7B | | 4B | | 8B | | 14B | | 32B | |
| | | PPL ↓ | Acc ↑ | PPL ↓ | Acc ↑ | PPL ↓ | Acc ↑ | PPL ↓ | Acc ↑ | PPL ↓ | Acc ↑ |
| | Baseline | 16.68 | 53.33 | 13.64 | 58.97 | 9.71 | 62.04 | 8.64 | 65.28 | 7.59 | 45.02 |
| 0.9 | SVD-LLM | 30.99 | 44.18 | 17.15 | 49.28 | 12.52 | 55.96 | 11.07 | 58.82 | 9.23 | 43.23 |
| | +LEMS | **17.62** | **51.24** | **14.02** | **57.56** | **9.83** | **62.36** | **8.77** | **65.28** | **7.64** | **65.61** |
| | KFAC-SVD | 25.89 | 42.94 | 16.01 | 50.67 | 11.51 | 57.07 | 9.96 | 60.00 | 8.50 | 43.24 |
| | +LEMS | **16.68** | **51.91** | **13.91** | **57.99** | **9.85** | **62.89** | **8.79** | **65.22** | **7.64** | **65.73** |
| 0.8 | SVD-LLM | 36.73 | 38.75 | 19.93 | 45.76 | 14.05 | 52.31 | 12.16 | 56.54 | 10.22 | 57.87 |
| | +LEMS | **18.72** | **46.24** | **15.13** | **54.29** | **10.50** | **60.04** | **9.32** | **62.85** | **7.90** | **65.85** |
| | KFAC-SVD | 29.45 | 40.30 | 17.55 | 46.38 | 12.52 | 53.56 | 10.73 | 56.02 | 9.20 | 42.21 |
| | +LEMS | **17.83** | **45.91** | **14.72** | **54.79** | **10.38** | **59.28** | **9.25** | **62.55** | **7.88** | **65.81** |
| 0.7 | SVD-LLM | 57.67 | 35.40 | 28.72 | 38.53 | 17.11 | 45.76 | 14.75 | 52.49 | 11.65 | 54.78 |
| | +LEMS | **23.25** | **42.05** | **17.11** | **48.91** | **12.37** | **54.75** | **10.40** | **58.32** | **8.34** | **64.51** |
| | KFAC-SVD | 39.56 | 36.60 | 21.51 | 41.50 | 14.69 | 46.78 | 12.21 | 52.03 | 10.26 | 55.39 |
| | +LEMS | **21.51** | **40.70** | **16.14** | **50.75** | **11.81** | **53.80** | **10.24** | **57.60** | **8.34** | **64.63** |
| 0.6 | SVD-LLM | 124.98 | 33.25 | 59.27 | 33.87 | 28.77 | 36.81 | 19.81 | 43.29 | 15.43 | 48.60 |
| | +LEMS | **32.35** | **37.23** | **22.45** | **43.53** | **17.28** | **46.46** | **13.35** | **51.12** | **9.36** | **60.37** |
| | KFAC-SVD | 84.89 | 33.45 | 39.79 | 35.82 | 21.68 | 39.63 | 15.49 | 44.62 | 12.57 | 51.12 |
| | +LEMS | **28.83** | **37.04** | **20.24** | **43.68** | **15.70** | **45.51** | **12.57** | **50.90** | **9.34** | **60.41** |
| 0.5 | SVD-LLM | 227.19 | 32.05 | 159.22 | 32.19 | 65.60 | 33.58 | 31.85 | 36.30 | 25.79 | 39.37 |
| | +LEMS | **56.33** | **35.06** | **33.57** | **38.28** | **31.66** | **40.70** | **18.32** | **42.72** | **11.88** | **54.59** |
| | KFAC-SVD | 198.93 | 32.70 | 172.84 | 32.36 | 40.18 | 35.02 | 23.67 | 37.71 | 18.76 | 42.75 |
| | +LEMS | **44.13** | **34.53** | **29.98** | **37.62** | **23.57** | **38.65** | **17.21** | **42.73** | **11.69** | **53.96** |

