# OpenReview forum: "Advancing SVD-based LLM Compression via Layer-Wise Error Model Search"
_ICML.cc/2026/Conference — ICML 2026 regular_

### Official Review · Reviewer_r3Le · 2026-02-21

**Soundness:** 3
**Presentation:** 3
**Significance:** 2
**Originality:** 2
**Overall Recommendation:** 5
**Confidence:** 4

**Summary:**

This work proposes a low-rank approach for LLM compression based on Kronecker Factorized Fisher approximation and dedicated layerwise sensitivity scoring, comprising both local and global saliency. The method is evaluated on a compression of a family of open-source LLMs, such as Qwen and Llama.

**Compliance With Llm Reviewing Policy:**

Affirmed.

**Final Justification:**

After reading the rebuttal and responses to other reviewers I decided to raise the score.

Overall approach is sound and achieves pretty decent result on model compression while being computationally tangible.

**Key Questions For Authors:**

* How long does the whole procedure take for larger LLM? For instance, Llama-3-70b.

**Limitations:**

-

**Strengths And Weaknesses:**

Strengths

- The proposed LEMS sensitivity score, accounting for local and global sensitivity, appears to be novel and noticeably improves perplexity/accuracy as compared to uniform and non-uniform baselines.
- K-FAC SVD, especially coupled with LEMS, outperforms prior state-of-the-art at different compression ratios. The adopted K-FAC approximation, despite simple, turns out to be more robust to noise compared to competitive Fisher SVD approaches.
- The importance of each method component is ablated and the breakdown into execution time / performance gain is provided.

Weaknesses
- Some baselines seem to be poorly tuned. GFWSVD is expected to achieve scores competitive or better to ASVD/SVD-LLM baselines.
- Basis Sharing method, despite being included in the related work, is not present in the comparison. I would suggest adding it.

---

> ### Author Rebuttal · Authors · 2026-03-30
>
> **We express our gratitude to the reviewers for their careful reading and critical assessment of our work.** We are greatly encouraged by the overall positive reception of our work, reflected in the reviewers' consensus across most reviewing criteria. We are glad to hear they found our approach of SVD-based compression and global rank allocation 'meaningful and technically relevant' (xsWA), 'practically important', 'novel' (r3le) and 'well written' (RGs5). Moreover, we are pleased that the reviewers acknowledged our efforts to evaluate and ablate our approach well (xsWa, r3le), and to show clear 'improvements on previous methods' (RGs5). We will address your questions and comments below and incorporate your feedback accordingly.
> ### Q1) Execution Time for Large LLM
> We provide the execution runtime details for Llama-3.1-70B in the supplement, Tab.4 Section B.4. The decomposition time is consistent across the search methods shown and is approximately 11h. Since the initial submission, we have further optimized the search implementation, lowering the cost of the activation-based searches to 15.2h for ASVD and 4.5h for LEMS. We emphasize that this is a one-time, offline cost, and the computed sensitivity metrics are fully cacheable for reuse across different compression targets. While energy-based searches incur almost no overhead, their downstream performance significantly trails activation-based methods at this scale.
> ### W1) Tuning of Baselines
> We appreciate the thorough review of the baselines. We devoted significant effort to ensuring that the provided baselines are carefully tuned. The apparent **discrepancy arises because our evaluation framework strictly isolates the decomposition method from the rank allocation** search to ensure a fair comparison. SVD-LLM is a highly competitive decomposition baseline, but its authors relied purely on uniform rank allocation. Other prior works (e.g., GFWSVD) enhance their results by reporting their decomposition with search against SVD-LLM without search.
> When evaluated using the exact same uniform baseline, the performance aligns with our reports. To demonstrate this, we re-ran GFWSVD utilizing the ASVD search used in their paper on Llama-2-7B. Llama-2 is referenced in GFWSVD’s paper and supported in the official repo, which served as close reference for our code. As shown below, our implementation aligns with their reports and confirms that the use of ASVD results in a close match to SVD-LLM in perplexity.
> |Approach|wiki|piqa|openqa|hellas|arc_c|arc_e|wino|
> |---|---|---|---|---|---|---|---|
> |KFAC-SVD (uniform)|7.53|0.69|0.28|0.41|0.31|0.63|0.64|
> |SVD-LLM (uniform)|8.42|0.67|0.26|0.40|0.27|0.57|0.63|
> |GFWSVD (uniform)|159.85|0.56|0.15|0.28|0.20|0.28|0.51|
> |GFWSVD (with ASVD)|8.74|0.73|0.25|0.46|0.32|0.66|0.64|
> ### W2) Missing BasisSharing Comparison
> You are correct to point out that we should have specifically stated that we opt to strictly compare against SVD approaches using the same base decomposition structure. BasisSharing presents a unique memory-computation trade-off: it maintains higher intermediate ranks at the same disk parameter count because sharing saves storage (Wang et al. 25a). However, during execution, the shared basis has higher intermediate ranks, which increases FLOPs at the same parameter count. Because of this conceptual difference, to ensure a fair comparison, we did not compare to them in the main text.
>
> Here, we add the comparison, comparing performance for matched memory and compute budgets on **Llama-2-7B** using values from their paper. At matched memory, **our method is very competitive** in perplexity, slightly lower in accuracy, while **at matched compute we significantly outperform BasisSharing** (both uniform compressed).
> Moreover, LEMS is directly compatible with Basis Sharing by adding a single constraint forcing shared layers to pick the same rank. In our PoC below, we **show that applying LEMS to BasisSharing improves performance**.
>
> **0.8 Compute Budget**
> |Approach|GParams|TFLOPs|wiki|c4|piqa|openqa|hellas|arc_c|arc_e|wino|
> |---|---|---|---|---|---|---|---|---|---|---|
> |BasisS|4.53|21.40|9.69|23.86|0.66|0.26|0.38|0.27|0.58|0.62|
> |KFAC|5.18|21.22|*7.53*|*15.43*|*0.69*|*0.28*|*0.41*|*0.31*|*0.63*|*0.64*|
>
> **0.8 Memory Budget**
> |Approach|GParams|TFLOPs|wiki|c4|piqa|openqa|hellas|arc_c|arc_e|wino|
> |---|---|---|---|---|---|---|---|---|---|---|
> |BasisS|5.18|24.46|7.77|*15.30*|*0.70*|0.27|*0.43*|*0.33*|*0.66*|0.63|
> |KFAC|5.18|21.22|*7.53*|15.43|0.69|*0.28*|0.41|0.31|0.63|*0.64*|
>
> **Using LEMS with Basis Sharing**
> |Approach|GParams|wiki|c4|piqa|openqa|hellas|arc_c|arc_e|wino|
> |---|---|---|---|---|---|---|---|---|---|
> |BasisS (paper)|5.18|7.77|15.30|0.70|0.27|0.43|0.33|0.66|0.63|
> |BasisS (our impl.)|5.18|8.12|19.28|0.68|0.26|0.42|0.26|0.56|0.63|
> |BasisS (our impl.)+**LEMS**|5.18|*6.26*|*9.87*|*0.75*|*0.31*|*0.52*|*0.38*|*0.72*|*0.67*|
>
> We thank you for your time and consideration and welcome further discussion if open questions remain.

---

> > ### Author Rebuttal · Reviewer_r3Le · 2026-04-01
> >
> > My concerns were adequately addressed, therefore I decided to increase my score.

---

### Official Review · Reviewer_RGs5 · 2026-03-09

**Soundness:** 3
**Presentation:** 4
**Significance:** 3
**Originality:** 3
**Overall Recommendation:** 5
**Confidence:** 4

**Summary:**

The work improves SVD‑based LLM compression by making better layer‑wise rank decisions under a global compression rate. It proposes a new loss proxy and uses a calibration dataset together with an ILP formulation to allocate ranks across layers based on this proxy. A Kronecker‑factor approximation of the Fisher Information, computed using token‑wise second moments, is then used to perform SVD‑based dimension reduction with the assigned ranks. The method demonstrates improved perplexity and QA performance compared with SOTA baselines at the same compression rate.

**Compliance With Llm Reviewing Policy:**

Affirmed.

**Key Questions For Authors:**

1. What about using an even smaller c_ref than 0.1? Would the performance improve further?
2. Have you considered using a shared low‑rank basis across heads or layers? I am curious whether this would affect performance.
3. Are there some specific failure cases for the proposed method?

**Limitations:**

yes

**Strengths And Weaknesses:**

Strengths:
1. The paper is very well written, it is a pleasure to read, claims is well supported by experiments.
2. The use of ILP to quickly find a global rank allocation, given a target compression rate across layers, is a clear improvement over previous methods.
3. The performance surpasses the previous SOTA under the same compression settings.

Weaknesses:
1. The method requires calibration data and is not applicable to MoE or state‑space LLMs.
2. Generalizability beyond standard QA tasks is not evaluated; it remains unclear whether the approach extends to specialized domains such as coding or math.
3. At the beginning of page 5, the transition from (i) to (a) is abrupt and confusing. Using italic formatting to indicate a subsubsection would improve readability.
4. KFAC‑SVD appears to be simply KFAC under a new name. It would be more appropriate to include it in the preliminaries rather than presenting it as part of the proposed methodology.

---

> ### Author Rebuttal · Authors · 2026-03-30
>
> **We express our gratitude to the reviewers for their careful reading and critical assessment of our work.** We are greatly encouraged by the overall positive reception of our work, reflected in the reviewers' consensus across most reviewing criteria. We are glad to hear they found our approach of SVD-based compression and global rank allocation 'meaningful and technically relevant' (xsWA), 'practically important', 'novel' (r3le) and 'well written' (RGs5). Moreover, we are pleased that the reviewers acknowledged our efforts to evaluate and ablate our approach well (xsWa, r3le), and to show clear 'improvements on previous methods' (RGs5). We will address your questions and comments below and incorporate your feedback accordingly.
>
> ### Q1) Lower $c_{ref}$ Values
> While we investigated larger $c_{ref}$ values in Fig. 3 / Sec. 5.3.3, upon your recommendation, we extended our ablation to explore the lower end, specifically $c_{ref}=0.01$ and $0.05$. The results are as follows:
> |C_ref|KL (Wiki)|KL (acc)|MSE (Wiki)|MSE (acc)|
> |---|---|---|---|---|
> |0.1 (paper)|14.38|54.8|14.52|54.0|
> |0.05|14.35|54.3|14.75|53.0|
> |0.01|14.30|54.5|14.55|53.5|
>
> Thus, extremely low values of $c_{ref}$ can slightly improve perplexity in the case of KL calibration, but come at the cost of zero-shot accuracy degradation.
> ### Q2) Shared Basis Across Layers
> We have considered using Shared Bases. However, due to its specific memory-computation trade-off (it increases active compute/FLOPs during inference), we decided not to pursue it as our primary focus. **Please see our detailed response W2 to reviewer r3Le** for empirical comparisons and how LEMS can be integrated with it.
> ### Q3) Failure Cases
> Across our studies, we have not observed concrete failure cases for the **LEMS** rank allocation. However, it could fail if a layer's sensitivity is inaccurately represented during calibration. For example, a severe underestimation of sensitivity could lead the solver to aggressively over-compress that layer (because the proxy cost appears artificially low), which could bottleneck the entire model.
> **KFAC-SVD** (and SVD-LLM), as discussed in Sec. 5.3.1, can sometimes suffer under very mild, uniform compression of large models. Truncating only a few eigenvalues leaves a large number of very small values, which can occasionally cause underflow or numerical instability under fp16 evaluation. Fortunately, global search methods like LEMS naturally prevent this behavior.
> ### W1) Applicability to MoE and SSMs
> We are unaware of a methodological limitation preventing the use of LEMS or KFAC-SVD on MoE or SSM architectures. For MoE models, our search can naturally incorporate expert routing frequencies to weight importance, provided the calibration data sufficiently triggers the relevant experts. For SSMs, each layer can be decomposed and assessed individually (identical to Transformers), making it fundamentally compatible. Because our error accumulation approximation is depth-dependent rather than micro-architecture-dependent, we expect it to transfer effectively.
> To test this, we compressed the **tiiuae/falcon-mamba-7b SSM** to 0.8 **using our approach without modification.** As seen below, KFAC improves over SVD-LLM, with LEMS further advancing performance. We believe further investigations on SSM and MoE are interesting directions future work.
> | Approach|wiki|ptb|c4|piqa|openqa|hellas|arc_c|arc_e|wino|mathqa |
> |---|---|---|---|---|---|---|---|---|---|---|
> | SVD-LLM|7.48|29.83|25.84|*0.71*|*0.31*|*0.44*|*0.38*|0.71|**0.69**|*0.29*|
> | KFAC-SVD|*7.18*|*23.94*|*24.10*|*0.71*|**0.32**|**0.45**|**0.41**|*0.74*|**0.69**|**0.31** |
> | KFAC-SVD + LEMS|**6.54**|**22.33**|**20.98**|**0.74**|**0.32**|**0.45**|**0.41**|**0.75**|*0.68*|**0.31** |
>
> ### W2) Domain Specific Knowledge
> To address this limitation, we extended the main evaluation to the MathQA dataset, commonly used in SVD compression. The accuracy below (ratio=0.8) aligns with the improvements reported on other benchmarks in the paper. Upon your recommendation, we will gladly include these results in the camera-ready version to confirm the method's generalizability to specialized reasoning domains.
> | MathQA at 0.8|mistral-7b|llama-3-8b |qwen3-8b |
> |---|---|---|---|
> |SVD-LLM|0.29|0.26|0.30|
> |KFAC-SVD|0.29|0.28|0.32|
> |+LEMS|0.29|0.29|0.38|
>
> ### W3) Formatting and KFAC-SVD Framing
> Thank you for pointing out the formatting issues on page 5. We will update the subsubsections to use italics. Regarding KFAC, we agree it could have been introduced in the preliminaries. We deliberately placed it in the core methodology to clearly contrast its properties with the Lanczos estimator (GFWSVD) and explain *why* it resolves the rank-bottleneck critical to Fisher-weighted SVD. We feel this aids understanding, but we will ensure it is transparent that the KFAC Fisher approximation itself is not our novel contribution.
>
> We thank you for your time and consideration and welcome further discussion if open questions remain.

---

> > ### Author Rebuttal · Reviewer_RGs5 · 2026-04-03
> >
> > The rebuttal adequately addresses my concerns, and I will keep my score unchanged.

---

### Official Review · Reviewer_xsWA · 2026-03-11

**Soundness:** 3
**Presentation:** 3
**Significance:** 3
**Originality:** 2
**Overall Recommendation:** 4
**Confidence:** 4

**Summary:**

This paper studies SVD-based compression for LLMs, focusing on two challenges: unstable Fisher-based decomposition and suboptimal layer-wise rank allocation. The authors propose KFAC-SVD, which uses token-level second-order statistics instead of sequence-level Fisher fitting to improve stability, and LEMS, which performs global rank allocation via a layer-wise error model and ILP. Experiments show improved perplexity and zero-shot accuracy over several baselines.

**Compliance With Llm Reviewing Policy:**

Affirmed.

**Key Questions For Authors:**

** Questions**:
1. Regarding cross-token second moments, the paper explicitly states that they are omitted in the token-wise approximation. Under what conditions is this omission expected to be benign, and under what architectures or tasks might it fail? My assessment of the method’s generality depends on this point.
2. Table 3 suggests that much of the gain in LEMS comes from calibration, while the benefit of energy + ILP alone is limited. Can the authors discuss more explicitly whether the main gains come from global optimization itself, or from additional empirical measurement?

If authors answer these questions well, I might raise my score.

**Limitations:**

yes

**Strengths And Weaknesses:**

**Strengths**:
1.  The paper clearly identifies two real bottlenecks in SVD compression: unstable Fisher-based decomposition and difficult global rank allocation. Splitting the solution into KFAC-SVD and LEMS makes the overall method easy to follow.
2. The technical design is sensible for large LLMs. Token-wise second-order statistics increase the effective sample size and improve estimator stability in practice.
3.  The empirical analysis is fairly solid. The supplementary evidence on spectral collapse and held-out Fisher error supports the practical value of KFAC-SVD.
4. The problem is practically important, and the gains are shown on multiple 8B-scale model families. That makes the method look more general than model-specific.
5. The novelty mainly comes from combining existing ideas in a useful way rather than introducing a new theory. Even so, the recombination is meaningful and technically relevant.

**Weaknesses**:
1. The paper is not very rigorous theoretically. KFAC-SVD seems more stable, but it is not clearly shown to be a more faithful estimate of the true Fisher matrix.
2. LEMS relies on a multiplicative cost model with limited theoretical justification. Treating layer depth as the main source of error propagation feels too simplistic for Transformers.
3. The presentation could be more cautious about what LEMS achieves. It is better described as a proxy-driven heuristic than as a principled global model.

---

> ### Author Rebuttal · Authors · 2026-03-30
>
> **We express our gratitude to the reviewers for their careful reading and critical assessment of our work.** We are greatly encouraged by the overall positive reception of our work, reflected in the reviewers' consensus across most reviewing criteria. We are glad to hear they found our approach of SVD-based compression and global rank allocation 'meaningful and technically relevant' (xsWA), 'practically important', 'novel' (r3le) and 'well written' (RGs5). Moreover, we are pleased that the reviewers acknowledged our efforts to evaluate and ablate our approach well (xsWa, r3le), and to show clear 'improvements on previous methods' (RGs5). We will address your questions and comments below and incorporate your feedback accordingly.
> ### Q1) Omission of Cross-Token Moments
> In this work, **we propose LEMS**, which allocates ranks across decomposition methods, compression budgets, and model architectures, **and KFAC-SVD**, a specific method for improving Fisher-weighted SVD. Since LEMS is modular and not tied to a single decomposition backbone (Tab.14, response to W2 (r3Le)), in our opinion, this **question mainly concerns the local Fisher estimator used by KFAC-SVD** rather than the generality of the overall framework.
>
> In the LLM compression regime we study, the dominant limitation of Fisher-weighted SVD is the rank/conditioning bottleneck of coupled sequence-level Fisher estimation. We resolve it by applying KFAC, for which the loss of **cross-token structure** is not an intended omission but a **consequential effect**.
>
> The effectiveness of the better conditioning empirically manifests in a major improvement over GFWSVD (Tab.2). To isolate improved conditioning (**our main goal**) from the effect of modeling cross terms, we additionally test the Shampoo$^2$ (Morwani et al. 2025) estimator in this rebuttal. Like KFAC, it avoids the conditioning bottleneck, but unlike KFAC it retains some cross-token structure. Under identical settings, it performs worse than KFAC (see table), supporting that **omission of cross-terms is benign in the LLM setting** we study.
> |ppl (acc.)|Llama-3-8b|Qwen3-8b|Mistral-7b|
> |---|---|---|---|
> |Shampoo$^2$|13.9 (0.44)|13.8 (0.50)|7.6 (*0.49*)|
> |KFAC|*11.4* (*0.45*)|*12.5* (*0.51*)|*7.1* (*0.49*)|
>
>
> More broadly, in domains with heavier spatial/positional structure (e.g., vision), preserving cross-token terms **may** matter more. However, even there, reverting to coupled estimators (e.g. GFWSVD) is unlikely to be sufficient, since they retain the same conditioning bottleneck. This motivates future work investigating other estimators like Shampoo$^2$ for Fisher-SVD in such domains.
> ### Q2) Global Optimization vs. Empirical Measurement
> Indeed, using ILP with energy alone is limited *because of* its global optimization. The ILP solver optimizes any given input to global *proxy optimality*. However, because raw spectral energy lacks global calibration, relying on it as sole proxy for global optimization yields suboptimal task performance (Table 3). By calibrating the energy scores with a sensitivity measurement, we ensure the proxy accurately reflects the layer's true *global* impact. With proper impact scoring, the global, proxy-optimal solution found is significantly better at preserving task-relevant components. Therefore, the effectiveness of the global ILP optimization and the use of measurements are inherently tied together. Notably, ASVD uses 5 measurements per layer (vs. our 1) yet fails to match LEMS, showing that measurements alone are insufficient without a sound global allocation strategy.
> ### W1) Faithful Fisher Estimate
> Thank you for raising this point, we agree our wording should be more precise. KFAC-SVD does not aim to match the empirical Fisher more faithfully in cosine/energy, but to improve compression. As discussed in Q1, Lanczos/GFWSVD suffers from a rank bottleneck that makes the Kronecker factors ill-conditioned for the inversion required by Eq. 4, which results in degraded performance of Fisher-SVD. To quantify the compression effect directly, Supplement B.2 derives a generalization metric measuring Fisher loss after compression, showing that KFAC-SVD better preserves the Fisher information through its truncation, which is the core goal of this work.
> ### W2) Simplicity of the LEMS Cost Model
> While we agree that our depth-based cost model is simplistic, this is deliberate and prioritizes scalability: Because layer sensitivities vary drastically between models (e.g., Llama-3 vs. Qwen), complex error propagation models risk overfitting to specific micro-architectures. Our combined cost model avoids this issue, enabling robust scaling across diverse sizes and architectures. We underline this point by extending it to an SSM (see response W1 to Reviewer RGs5).
> ### W3) Presentation of LEMS
> We value your suggestion and will revise the wording (assuming it refers to l.120?)
>
>
> We thank you for your time and consideration and welcome further discussion if open questions remain.

---

> > ### Author Rebuttal · Reviewer_xsWA · 2026-04-03
> >
> > I find the rebuttal helpful and it resolves a substantial part of my concerns, especially on the KFAC-SVD side. For LEMS, however, I still think the paper would benefit from a clearer attribution of where the observed gains come from: better proxy calibration, better global optimization, or their interaction. I will maintain my positive score.

---

> > > ### Author Response · Authors · 2026-04-04
> > >
> > > Thank you again for the additional feedback. We are glad that our previous response resolved your concerns regarding KFAC-SVD.
> > >
> > > To further clarify the individual contributions of proxy calibration and global optimization in LEMS, we ran a **cross-ablation** on **Llama-3-8B @ 0.8**, combining each score-based search (MRCS / ASVD / LEMS) with each of their proxy scores (Energy / KL / Hybrid / Hybrid+bias).
> > >
> > > **TLDR:** Neither calibration nor optimization alone explain the gains observed with LEMS. Instead, its performance gains can be attributed to *the interaction* of our calibrated proxy (Eq.11) and global optimization formulation (Eq.15).
> > >
> > > -------
> > >
> > > ## Cross-Ablation
> > > The evaluation of the cross-ablation reveals the following (results in table below):
> > >
> > > **1. Global optimization alone is inconsistent without a calibrated proxy.**
> > > When applied to an uncalibrated proxy like energy, LEMS's global solver does not guarantee improvement (LEMS 14.58, MRCS 11.83 and ASVD 10.08). This demonstrates that a global optimizer can be easily misled if the underlying proxy metric does not accurately reflect true global layer importance. When using the KL proxy, the global solver's performance is competitive, though gains are minor.
> > >
> > > **2. Proxy calibration alone is constrained by search.**
> > > Pairing our hybrid+bias proxy with prior search algorithms can improve the performance (e.g., MRCS from 11.83 to 11.33 ppl), but, compared to other proxies, only yields minor gains. This indicates that simply *providing a better-calibrated, dense proxy*, is *not guaranteed to yield significant improvements*, even when additional measurements are used for calibrating the bias.
> > >
> > > **3. Interaction: Calibration and optimization must be aligned.**
> > > The largest gains arise when the calibrated proxy and the global optimizer are aligned. The combination of our hybrid+bias proxy and ILP formulation pushes performance significantly beyond prior search baselines (regardless of proxy), reaching *8.23 ppl and 0.52 accuracy*. **Hence, LEMS's strong performance can be attributed to its interaction between the calibrated proxy and the global optimization.**
> > >
> > > **Llama-3-8B (0.8 remaining params)**
> > > *Metrics: WikiText2 perplexity (Avrg. zero-shot accuracy), best search per score in **bold**.*
> > > | Proxy Score | Score Type | MRCS search | ASVD search | LEMS search |
> > > | :--- | :--- | :--- | :--- | :--- |
> > > | Energy (MRCS) | Dense | 11.83 (0.44) | **10.08** (0.47) | 14.58 (0.47) |
> > > | KL (ASVD) | Sparse | n.a. | **11.01** (0.45) | 11.53 (0.46) |
> > > | Hybrid (ours) | Dense | 11.22 (0.45) | 12.16 (0.45) | **8.74** (0.49) |
> > > | Hybrid + bias (ours) | Dense | 11.33 (0.47) | 10.28 (0.49) | **8.23** (0.52) |
> > >
> > > *(Note: MRCS cannot operate on the KL metric because it requires a dense sensitivity score, whereas ASVD's KL is sparse.)*
> > >
> > > ## Extended Results
> > >
> > > We additionally **repeated this cross-ablation on Qwen3-8B and Mistral-7B** (and also at 0.6 retention, omitted here for space). These results support the **same overall conclusion**: The extended results **align with the observed trends** in the cross-ablation of Llama: the best perplexity and overall tradeoff are typically achieved when the calibrated dense proxy of LEMS is paired with its global ILP formulation, although the relative contribution of proxy and search varies somewhat across architectures.
> > > The full ablation will be included in the CR version.
> > >
> > > **Qwen3-8B (0.8)** *WikiText2 Perplexity (Avrg. zero-shot Acc.)*
> > > | Proxy Score | Score Type | MRCS search | ASVD search | LEMS search |
> > > | :--- | :--- | :--- | :--- | :--- |
> > > | Energy (MRCS) | Dense | 14.27 (0.48) | 14.58 (0.54) | **12.52** (0.51) |
> > > | KL (ASVD) | Sparse | n.a. | 11.53 (0.56) | **11.29** (0.57) |
> > > | Hybrid+bias (ours)| Dense | 12.62 (0.51) | 12.04 (0.53) | **10.40** (0.56) |
> > >
> > > **Mistral-7B (0.8)** *WikiText2 Perplexity (Avrg. zero-shot Acc.)*
> > > | Proxy Score | Score Type | MRCS search | ASVD search | LEMS search |
> > > | :--- | :--- | :--- | :--- | :--- |
> > > | Energy (MRCS) | Dense | 7.03 (0.49) | **6.64** (0.51) | 10.26 (0.46) |
> > > | KL (ASVD) | Sparse | n.a. | 6.85 (0.51) | **6.39** (0.51) |
> > > | Hybrid+bias (ours)| Dense | 7.02 (0.49) | 6.74 (0.49) | **5.97** (0.54) |
> > >
> > >
> > > Thank you for your continued effort in reviewing. We hope this additional analysis clarifies your question!

---

### Decision · Program_Chairs · 2026-04-30

**Decision:**

Accept (regular)

**Comment:**

This paper studies low-rank SVD-based compression for LLMs and addresses two key limitations of prior methods: ineffective global rank allocation and poor decomposition quality caused by rank collapse in Fisher-based estimators. To tackle these issues, the authors propose LEMS, a layer-wise error modeling approach that captures local/global importance and propagation bias for efficient global rank search via ILP, and KFAC-SVD, which leverages token-wise statistics to improve decomposition quality and avoid rank deficiency. Experiments on the Mistral, Qwen3, and Llama-3 families show consistent gains in perplexity and zero-shot accuracy, with improvements that generalize up to 70B-scale models.

Overall, the reviewers agree that this paper makes solid and novel contributions. I therefore recommend accepting this paper.